# Recent Advances in the Digestive, Metabolic and Therapeutic Effects of Farnesoid X Receptor and Fibroblast Growth Factor 19: From Cholesterol to Bile Acid Signaling

**DOI:** 10.3390/nu14234950

**Published:** 2022-11-22

**Authors:** Agostino Di Ciaula, Leonilde Bonfrate, Jacek Baj, Mohamad Khalil, Gabriella Garruti, Frans Stellaard, Helen H. Wang, David Q.-H. Wang, Piero Portincasa

**Affiliations:** 1Clinica Medica “A. Murri”, Department of Biomedical Sciences & Human Oncology, University of Bari “Aldo Moro” Medical School, 70124 Bari, Italy; 2Department of Anatomy, Medical University of Lublin, 20-059 Lublin, Poland; 3Section of Endocrinology, Department of Emergency and Organ Transplantations, University of Bari “Aldo Moro” Medical School, 70124 Bari, Italy; 4Institute of Clinical Chemistry and Clinical Pharmacology, Venusberg-Campus 1, University Hospital Bonn, 53127 Bonn, Germany; 5Department of Medicine and Genetics, Division of Gastroenterology and Liver Diseases, Marion Bessin Liver Research Center, Einstein-Mount Sinai Diabetes Research Center, Albert Einstein College of Medicine, Bronx, NY 10461, USA

**Keywords:** bile acids, nonalcoholic fatty liver disease, FGF15/19, FXR, agonist, nuclear receptors

## Abstract

Bile acids (BA) are amphiphilic molecules synthesized in the liver (primary BA) starting from cholesterol. In the small intestine, BA act as strong detergents for emulsification, solubilization and absorption of dietary fat, cholesterol, and lipid-soluble vitamins. Primary BA escaping the active ileal re-absorption undergo the microbiota-dependent biotransformation to secondary BA in the colon, and passive diffusion into the portal vein towards the liver. BA also act as signaling molecules able to play a systemic role in a variety of metabolic functions, mainly through the activation of nuclear and membrane-associated receptors in the intestine, gallbladder, and liver. BA homeostasis is tightly controlled by a complex interplay with the nuclear receptor farnesoid X receptor (FXR), the enterokine hormone fibroblast growth factor 15 (FGF15) or the human ortholog FGF19 (FGF19). Circulating FGF19 to the FGFR4/β-Klotho receptor causes smooth muscle relaxation and refilling of the gallbladder. In the liver the binding activates the FXR-small heterodimer partner (SHP) pathway. This step suppresses the unnecessary BA synthesis and promotes the continuous enterohepatic circulation of BAs. Besides BA homeostasis, the BA-FXR-FGF19 axis governs several metabolic processes, hepatic protein, and glycogen synthesis, without inducing lipogenesis. These pathways can be disrupted in cholestasis, nonalcoholic fatty liver disease, and hepatocellular carcinoma. Thus, targeting FXR activity can represent a novel therapeutic approach for the prevention and the treatment of liver and metabolic diseases.

## 1. Introduction

Bile acids (BA) are components of human bile, a brownish or olive-green liquid made of water, organic solutes, and inorganic salt, at a pH of 7.6–8.6. Bile is the major excretion route of cholesterol which appears as solubilized cholesterol and as BA, i.e., degradation products. The three main lipids in bile are bile salts, phospholipids, and nonesterified cholesterol. Free cholesterol accounts for 97% of all sterols in bile, the rest are cholesterol precursors and dietary phytosterols. BA are amphipathic molecules synthesized as “primary” BA from cholesterol in the pericentral hepatocytes. Following their synthesis, primary BA are conjugated with the compound taurine (2-aminoethanesulfonic acid) and the amino acid glycine; this step increases BA solubility in aqueous solutions, such as bile. Primary-conjugated BA are actively secreted from the hepatocyte into bile and are mainly and actively absorbed in the terminal ileum. A small amount will travel to the colon and will undergo the microbiota-dependent biotransformation into “secondary” and “tertiary” BA, which will be passively absorbed. By this dynamic mechanism, BA undergo continuous enterohepatic circulation with 4–12 cycles/day. The fecal loss is minimal at every cycle (5%) and must be compensated by the de novo synthesis of primary BA in the liver (~200–600 mg/daily). The synthesis of BA is modulated by negative feedback mechanisms controlled by the farnesoid X receptor (FXR) in the liver and in the intestine [1,2]. This pathway is strictly linked with the enterohepatic circulation of BA [3], with the signaling role of BA [4], and with the composition and abundance of gut microbiota [5].

Because of this scenario, the role of BA is not simply limited in the emulsification and absorption of dietary fat and fat-soluble vitamins. BA also regulate the proliferation of epithelial cells [6,7], gene expression [8,9], epigenetic mechanisms [10,11], fibrogenesis [12], lipid [13] and glucose metabolism [14]. These effects derive from the role of BA as endogenous ligands and from their ability to activate specific receptors. Besides FXR, involved receptors also include the membrane-associated G-protein-coupled bile acid receptor-1 (GPBAR-1, also known as TGR5 or M-BAR) [6,15], and sphingosine-1-phosphate receptor 2 (S1PR2) [16,17] in the intestine, in the liver, in the muscle and in the brown adipose tissue [18,19]. In the latter decades, the recognition of BA as signaling molecules has shed new light on the complex pathophysiological mechanisms and potential therapeutic implications. Consequences of disrupted BA homeostasis include cholestasis [20], hepatic steatosis, liver fibrosis, and liver tumor [21,22]. Notably, therapeutic manipulation of the BA-FXR axis is paving the way to innovative and potent therapeutic tools [3,23,24], potentially able to act at multiple levels (i.e., BA composition, mitochondrial function, modulation of gut microbiota, glucose and lipid homeostasis, liver inflammation) [25,26,27,28,29,30,31,32,33]. The present review depicts the complex relationship linking BA and FXR, and how its modulation can become a valid therapeutic target for several liver diseases.

## 2. The Enterohepatic Circulation and Kinetics of BA and the FXR–FGF19 Dynamics

The cycling of BA between the liver and the intestine is defined enterohepatic circulation, while the total burden of BA in the enterohepatic circulation represents the circulating BA pool [34]. The complex process operating in the enterohepatic circulation of BA has been extensively described in previous papers [19,21]. Briefly, the enterohepatic circulation implies that BA reach the terminal ileum and the colon as well. In the ileum, BA act as signaling molecules and agonists to the nuclear receptor FXR. FXR has antimicrobial effects and protects the intestinal barrier. FXR activation is linked, in humans, with the transcription of the enterokine fibroblast growth factor 19 (FGF19) or the ortholog FGF15 in mice. FGF19 enters the portal flow and has effects on the gallbladder and liver [21]. BA activate FXR with the following rank order: chenodeoxycholic acid (CDCA) > lithocholic acid (LCA) = deoxycholic acid (DCA) > cholic acid (CA) in the conjugated and unconjugated forms [35]. 

FGF19 acts as agonist of the hepatic FGF receptor 4 (FGFR4)/β-Klotho and activates c-Jun *N*-terminal kinase/extracellular signal-regulated kinase (JNK/ERK). This pathway inhibits the expression of CYP7A1 and CYP8B1 and further hepatic BA synthesis, in concert with the FXR- small heterodimer partner (SHP) inhibitory pathway [36]. BA travelling from the intestine into the portal tract enter the liver via the sodium taurocholate cotransporting polypeptide (NTCP) and organic anion transporting polypeptide (OATP) transporters. FXR works with other nuclear receptors involved in BA homeostasis, namely retinoid X receptor (RXR), SHP, liver receptor homologous-1 (LRH-1), and liver X receptor (LXR) [18]. FXR is the main sensor of BA and regulates synthesis, secretion, and metabolism of BA in the liver, ileum and colon [36,37,38]. In the liver, FXR regulates target gene transcription by binding to RXRs as a heterodimer [39]. This binding leads to increased transcription of the SHP expression. SHP, in turn, inhibits CYP7A1 expression by blocking transactivation of the hepatic activators LRH-1 and hepatic nuclear factor 4 (HNF-4), at the promoter [39]. This pathway ultimately prevents the activation of target genes involved in the synthesis of fatty acids and BA. By contrast, at low concentrations of BA, LRH-1 operates together with LXR and stimulates the synthesis of BA [40,41,42]. FXR transcriptionally activates the enzymes involved in BA conjugation to glycine or taurine (bile acid-CoA synthetase [BACS] and bile acid-CoA: amino acid *N*-acetyltransferase [BAT]) [43], regulates hepatic BA secretion by bile salt export pump (BSEP), and hepatic phospholipid secretion by ABCB4. BA re-entering the liver also interact with GPBAR-1 on the macrophagic Kupffer cells, together with the pathway stimulated by the FGFR4/β-Klotho dimer. In addition, the activation of FXR induces expression of BA detoxification enzymes (i.e., cytosolic sulfotransferase 2A1 (SULT2A1 [44]), aldol-keto reductase 1 B7 (AKR1B7 [45]), cytochrome P450 3A4/3a11 (CYP3A4/Cyp3a11), and UDP-glycosyltransferase 2B4 (UTG2B4)) [46]. In the kidney (proximal tubule), circulating BA undergo uptake by the apical sodium/dependent bile acid transporter (ASBT). Glomerular filtration of BA are modulated by MRP 2, 3, 4 transporters [47].

Experimental models in animals show that the pathogenesis of NASH is likely linked with a markedly altered composition of BA in the enterohepatic- rather than in the systemic circulation, with specific reduction of secondary BAs (mainly DCA), and scarce presence of FXR and TGR5 ligands in the portal blood. In this case, the major role of enterohepatic BA in the pathogenesis of NASH is confirmed by the protection from NASH secondary to dietary correction (i.e., DCA supplementation) of the BA profile [48].

## 3. Regulation of BA Homeostasis: The Role of Gut Microbiota

The gut represents a dynamic interface between the internal and the external body environment [49], with several stimuli continuously interacting with the intestinal barrier [50]. The gut microbiota is sensitive to dietary habits and nutrients such as dietary fiber, proteins, fat, carbohydrates, [51]. They are also sensitive to external toxic agents, which include smoking [52], ethanol consumption [53], and environmental pollutants such as heavy metals and pesticides [54,55,56,57]. All these factors can affect the diversity and relative abundance of the gut microbiota [58] during the process of enterohepatic circulation, when the primary BA synthetized and secreted by the liver are transformed into secondary BA in the colon [58,59]. BA and gut microbes interact within a continuous bidirectional crosstalk in health but also in disease [50,60,61]. Gut dysbiosis can disrupt BA homeostasis and can change the composition of the BA pool, while the increased production of deoxycholic acid (DCA), a cytotoxic secondary BA, can damage the composition of gut microbiome [9]. BA bind to FXR and this step produces antimicrobial peptides (AMPs) (i.e., angiogenin 1 and RNase family member 4). These AMPs play an active role in the inhibition of gut microbial overgrowth and intestinal barrier dysfunction [62]. BA can also modulate the gut microbiome by stimulating the growth of BA-metabolizing bacteria or inhibiting other bile-sensitive bacteria. Gut *Eubacterium lentum, Ruminococcus gnavus* and *Clostridium perfringens* decrease the antimicrobial effect of BA via the iso-BA pathway by transforming DCA and LCA into iso-DCA and iso-LCA (3b-OH epimers). The secondary DCA has hydrophobic and cytotoxic profiles with detergent effects on the bacterial cell membranes and antimicrobial properties [63]. 

Changes of the microbiota and therefore BA pool can interfere with activation of signaling pathways [64] involved in intestinal, metabolic homeostasis and tumorigenesis. Dietary changes associated with low short-chain fatty acids have been linked with an high risk for cancer development [65,66]. Colorectal cancer represents another example since, in humans, dietary habits can vary BA conjugation. For example, diets enriched in animal protein favor taurine conjugation while vegetarian diets favor glycine conjugation. In this scenario, TCA can to stimulate gut microbes able to convert taurine and cholic acid to hydrogen sulfide and DCA, which act as genotoxin and tumor-promoter, respectively [65].

Dysbiosis and variations of the BA pool composition and size can play a role in alcohol associated liver disease (ALD) [67], in NAFLD, obesity, and type 2 diabetes [58,68,69,70]. Compared to healthy controls, higher levels of total faecal BA, primary CA and CDCA, and higher BA synthesis have been reported in patients with NASH, who also showed a higher ratio of primary to secondary BA, but a similar ratio of conjugated to unconjugated BA. Patients with NASH were also characterized by a decreased count of *Bacteroidetes* and *Clostridium leptum.* The count of *C. leptum* increased with fecal unconjugated LCA and decreased with unconjugated CA and CDCA. Taken together, these findings point to a link between NAFLD, dysbiosis and altered BA homeostasis, which puts patients at risk of further hepatic injury [71]. 

Dysbiosis can be also implicated in tumorigenesis via BA-induced changes. DCA increases on a western diet and becomes a predisposing factor to intestinal carcinogenesis. In DCA-treated APC (min/+) mice, a disrupted intestinal microbiota was associated with altered gut barrier, low-grade gut inflammation and tumor progression. Fecal microbiota transplantation from DCA-treated mice to Apc (min/+) mice increased tumor multiplicity, caused inflammation, recruited the M2 phenotype tumor-associated macrophages, and activated the tumor-associated Wnt/beta-catenin signaling pathway. Notably, the antibiotic-induced depletion of the microbiota blocked DCA-induced intestinal carcinogenesis [72]. Another important study found that a 2-week food exchanges profoundly affected the microbiota, metabolome profile, mucosal biomarkers of cancer risk when African Americans were fed a high-fibre, low-fat African-style diet and rural Africans were fed a high-fat, low-fibre western-style diet. In the African Americans the study found a protective profile by saccharolytic fermentation and butyrogenesis and suppressed secondary BA synthesis [66].

## 4. The Microbiota–BA–FXR Axis

The BA-FXR axis protects the liver for BA overload and potential harmful effects of BA upon accumulation in the hepato-biliary-intestinal tract [73]. Additional aspects of such interaction include systemic metabolic effects deriving from FXR activation and FGF19 secretion [19]. Thus, changes in the profile of the BA pool and primary/secondary BA ratio can produce multiple consequences, including altered metabolic pathways and liver damage [71,74,75]. In this context, a major role is played by microbiota-induced deconjugation of BA. This is the case when the murine primary BA Tauro-beta-muricholic Acid (TβMCA), a natural occurring FXR antagonist, will be deconjugated by the microbiota. This step will alleviate FXR signaling in ileum leading, in the liver, to inhibition of BA synthesis [60,76]. In normal conditions with a functioning enterohepatic circulation of a physiological BA pool, the FXR-induced gene expression including *Ang1*, *iNos* and *Il18* in ileum, has antimicrobial action, enteral protection and inhibition of bacteria damage to the intestinal mucosa. In rodents, the biliary obstruction that causes small intestinal bacterial overgrowth (SIBO) can be reversed by administration of BA [77,78]. A study in the cholestatic model of mice also reported protection by a potent synthetic agonist of FXR [79]. Metabolic studies in mice show that the gut microbiota promotes diet-induced obesity via FXR signaling [62,75,76]. In an animal model of NAFLD induced by high-fat diet, the antibiotic treatment decreased BSH-encoding *Lactobacillus*, increased the synthesis of the FXR antagonist TβMCA, and improved insulin resistance and liver steatosis [80]. In humans with newly diagnosed type 2 diabetes, naïve treatment with metformin modified gut microbiota, decreasing *Bacteroides fragilis* and increasing the BA glycoursodeoxycholic acid (GUDCA) in the gut. These changes were paralleled by an inhibition of FXR signaling, pointing to GUDCA as an intestinal FXR antagonist able to improve metabolic dysfunction [81]. The inhibition of intestinal FXR is a key factor for the progression of NAFLD mediated by gut microbiome [75]. Mice fed a high milk- fat diet showed a shift in BA composition with increased TCA and expansion of *Bilophila wadsworthia*. This proteobacterium is recognized as a ‘‘bile-loving’’ microorganism associated to inflammatory bowel diseases [82,83]. 

In NAFLD rats, the administration of probiotics is able to significantly increase the expression of FXR, FGF15 mRNA, and protein in the liver, upregulating the diversity of gut microbiota, downregulating the abundance of pathogenic bacteria and finally alleviating NAFLD [26].

One note, changes in gut microbiota composition are possible following FXR/FGF19 modulation. In humans with biopsy-confirmed NASH, administration of the FGF19 analog aldafermin induced in the microbiota a dose-dependent enrichment in *Veillonella*, a rare commensal microbe which correlated with changes in serum bile acid profile, mainly in terms of decreased toxic BA [84].

Dietary lipids likely impact the gut microbial composition directly as a substrate or by shifting in BA composition [85]. The microbiota FXR signaling modulation was also assessed during the sub-ministration of the antioxidant Tempol, which resulted in a decrease of *Lactobacillus* and *Clostridium* (clusters IV and XIVa), accompanied by decreased BSH, accumulation of TβMCA and suppressed FXR signals [76]. Other nuclear receptors can also be involved during the interaction of microbiota with intestinal BA, such as the pregnane X receptor (PXR), vitamin D receptor (VDR), GPBAR-1 and Sphingosine-1-Phosphate Receptor 2 (S1PR2) [86,87,88]. 

## 5. Fibroblast Growth Factors FGF15 and Human Ortholog FGF19

Fibroblast growth factors FGF15 (rodent) and the human ortholog FGF19 are enterokines which play a major role in BA homeostasis and key metabolic functions, in concert with FXR [89]. The transcriptional regulation of *FGF19* involves the FXR during the enterohepatic circulation of BA [90,91]. Additional players include vitamin D receptor (VDR), pregnane X receptor (PXR), drugs, vitamins and cholesterol [92]. The sterol regulatory element-binding protein 2 (SREBP2) is a negative transcriptional regulator of *FGF19* [93]. FGFs belong to a family of at least 22 proteins with an effect on growth, development, and differentiation [90,94,95]. FGFs, currently grouped into six subfamilies, play an autocrine and paracrine role via activation of specific tyrosine kinase FGF receptors undergoing dimerization and activation of the cascade of intracellular signaling pathways [96]. The trio consisting of FGF15 (and the human ortholog FGF19), FGF21, and FGF23 act as circulating hormones [97]. FGF19 is the human protein encoded by the *FGF19* gene and has endocrine hormonal functions at a systemic level [90,98]. FGF19 was originally identified in the fetal brain and, with FGF15 only shares ~50% amino acid identity [99]. FGF15/19 are predominantly expressed in the small intestine, gallbladder, kidney, skin, cartilage, and brain [36,100,101]. 

Previous studies confirmed that BA or FXR agonists induced FXR responsive element (FXRE) activation and FGF19-dependent repression of CYP7A1 (and therefore BA synthesis) in human hepatocytes [102]. A further proof was that *Fgf15*-knockout mice and intestine-specific *Fxr*-knockout mice stimulated by FXR agonists could not repress CYP7A1 [103].

FGF19 displays two daily peaks (about 3pm and 9pm). The serum peak of FGF15/19 follows the increase of BA in the small intestine and increases about 1.5–2.0 h after the rise of postprandial serum levels of BA [104,105]. FGF19 is a small 25 kDA molecule with fasting levels varying from 49 to 590 pg per mL. Following CDCA activation of the intestinal FXR, serum levels of FGF19 increase by +250%. FGF19 half-life is 30 min short, likely dependent on renal elimination [106]. FGF19 circulating levels can either decrease or increase in extrahepatic cholestasis, inflammatory bowel disease, kidney disease, BA malabsorption, obesity, and liver steatosis. Therapies involving FGF19 seem promising. However, the translational value of these attractive therapeutic tools should be carefully verified in terms of possible hepatic tumorigenesis, as a consequence of their mitogenic potential [107,108]. The effect likely involves the FGF19-FGFR4 interaction, since blocking the receptor can prevent, in rodents, the development of hepatocellular carcinoma [109].

The function of secreted FGFs requires the interaction with the transmembrane Klotho proteins found at the fibroblast growth factor receptors (FGFRs) [106]. FGF19 binds to the receptor complex composed of the FGF receptor 4 (FGFR4) and a co-receptor β-Klotho, both highly expressed in liver. Whereas the carboxy-terminal domain of FGF19 is the segment specifically recognized by β-Klotho, the amino-terminal region is specific for the FGF19–FGFR interaction [110].

Affinity of FGF15/19 is greater for the FGFR4, which is primarily expressed in the liver, than for FGFR1, primarily expressed in the white adipose tissue (WAT) [111,112]. FGFR4 is also expressed in other cell types as macrophages, HSCs and some central neurons [113]. The human FGF19 is not effective on mouse FGFR4 [114]. Binding of FGF19 to the FGFR4–β-Klotho complex activates a signaling pathway which includes the small guanosine triphosphatase Ras, the extracellular signal–regulated protein kinase 1, 2 (ERK1, ERK2), JUN *N*-terminal kinase (JNK), fibroblast growth factor receptor substrate 2α (FRS2α) [115], the growth factor receptor-bound protein 2 (GRB2), cAMP-response element-binding protein (CREB) which is de-phosphorylated and inactivated [89,106,116]. 

FGF19 is also able to modulate gallbladder volume, which contributes to the pulsatile and dynamic function of the enterohepatic circulation in the fasting and postprandial period [117,118]. During the postprandial period, the fat-stimulated CCK release from the upper gut enterocytes promotes smooth-muscle-mediated gallbladder contraction and ejection of concentrated bile into the duodenum. This phenomenon is well visible during the ultrasonographic functional study of time-dependent changes of gallbladder volume in response to a meal, by following the rhythmic alternation of emptying-refilling episodes [117,119,120,121]. Upon gallbladder contraction, BA in the duodenum inhibit further CCK production and, in turn, gallbladder contraction [122]. BA arriving in the distal ileum, moreover, will stimulate the release of FGF19 which, by activation of the gallbladder FGFR4-β-Klotho, promotes gallbladder relaxation ready for the next filling with dilute hepatic bile. FGF-19 interacts with the liver FGFR4-β-Klotho leading to suppression of the synthesis of hepatic BA [21]. 

## 6. FXR–FGF19 and BA Homeostasis

The hepatic and intestinal expression of FXR contributes to the regulation of BA synthesis and homeostasis [103], via a negative gut-liver feed-back sustained by the intestinal FGF15/19. In the liver, mild activation of FXR and SHP suppresses the CYP7A1/*Cyp7a1* gene [40,123]. In parallel, the *Cyp8b1* gene repression via FXR depends equally on both intestinal and liver FXR [103]. The key role of FXR in BA homeostasis is testified by the regulatory capacity of inducing the expression of key transporters active in the enterohepatic circulation of BA, i.e., BSEP, IBABP and OSTα/β, while suppressing NTCP and ASBT [124,125,126]. The BA-FXR-FGF15/19-BA sequence has a fundamental role in governing BA homeostasis. *FGFR4* overexpression downregulates CYP7A1 and decreases the BA pool size [127].

By contrast, Knockout mice *Fgf15*-, *Fgfr4*- and *β-klotho (Klb)*- display impaired BA metabolism. Exogenous FGF19 fails to rescue this condition and fails to repress the function in CYP7A1 *Fgfr4*- and *Klb*-knockout animals [128,129]. The interaction between FGF15/19 and β-klotho-FGFR4 in the liver requires additional nuclear receptors to activate CYP7A1 repression. These nuclear receptors include the liver receptor homologue 1 (LRH-1) and hepatocyte nuclear factor 4α (HNF4α) [130] and SHP [131]. In addition to this complex picture, BA bind to ileal membrane-associated receptor GPBAR1. This activation is also associated with several extra-intestinal tissues to mediate host energy expenditure [132,133], glucose homeostasis [134], anti-inflammatory and immune responses [135,136].

## 7. FXR–FGF15/19 Pathway and Metabolic Effects

The axis FXR–FGF15/19 is also implicated in a number of metabolic functions which involve fat, glucose, glycogen, protein homeostasis, and energy expenditure [106].

### 7.1. Lipid Homeostasis

The pathways connecting cholesterol to BA synthesis are essential and contribute to the prevention of pathological amounts of cholesterol in the body [137], and also to metabolic homeostasis. Hepatic BA synthesis starts from cholesterol, and this step represents the main catabolic pathway of cholesterol metabolism in humans [21]. These pathways act through the “classic” neutral pathway (cholesterol 7α-hydroxylase, CYP7A1), contributing to about 75% of the total BA pool, and the “alternative” acidic pathway (sterol 27-hydroxylase, CYP27A1) contributing to about 25% of the total BA pool [21]. 

In this context, FXR has a relevant role in atherosclerotic risk factors, and strongly modulates the homeostasis of cholesterol due to the ability to inhibit CYP7A1.

Of note, increased serum levels of primary and secondary BA have been reported in patients with NAFLD, as compared with healthy controls [9]. This finding should suggest a role for the elevated BA production in the pathogenesis of NAFLD. This link, however, seems secondary to the increased proportion, in the BA pool, of the FXR antagonistic DCA, and to decreased levels of the FXR agonistic CDCA [9]. On the other hand, the cholesterol catabolism through BA synthesis has beneficial effects on metabolic homeostasis, and hepatic accumulation of cholesterol is a critical culprit for the development of NAFLD/NASH [138]. Transgenic mice overexpressing CYP7A1 in the liver are resistant to high-fat diet-induced obesity, fatty liver and insulin resistance, mainly through increased hepatic cholesterol catabolism and increased BA pool [139]. Further studies in FXR-deficient and CYP7A1-deficient mice confirmed the beneficial effects on hepatic inflammation of the increased CYP7A1 expression and BA synthesis secondary to activation of FXR [27].

FXR-deficient mice display increased content of hepatic lipids and increased plasma cholesterol and triglycerides [140]. By contrast the activation of FXR by the synthetic agonist GW4064 or by BA decrease liver steatosis and serum triglycerides [141,142]. Mechanisms include suppression of de novo fatty acid synthesis [141]. Primary hepatocytes incubated with the recombinant FGF19 protein display suppression of insulin-dependent stimulation of fatty acid synthesis with or without insulin [143]. FXR activation induces the SHP-mediated suppression of sterol regulatory element-binding protein 1c (SREBP1c), a step decreasing triglyceride load in hepatocytes [141]. This pathway also occurs with FGF19 via increased expression of SHP, increased signal transducer and activator of transcription 3 (STAT3) and decreased peroxisome proliferator-activated receptor γ coactivator-1β (PGC-1 β) [143]. An additional mechanism is the FXR-mediated activation of ApoCII gene transcription in the liver. This step, in turn, activates the lipoprotein lipase and decreases ApoCIII, a lipoprotein lipase inhibitor. The net effect is increased lipolysis of triglycerides in the vasculature [30,144]. Additional effects of FXR activation include increased triglyceride hydrolysis and clearance, increased fatty acid oxidation [28,29,30], and decreased hepatic export of very low-density lipoprotein (VLDL) [145]. 

Negative-correlations have been recently shown between stimulated intestinal FXR-FGF15 secretion and pathways linking hepatic cAMP regulatory element-binding protein (CREB) and peroxisome proliferator-activated receptor gamma, coactivator 1α (PGC1A). This effect indicates a possible downregulation of hepatic PGC1α by inactivation of CREB, finally resulting in suppression of fatty acid oxidation [146].

FXR contributes to the control of cholesterol homeostasis. Human cells lack enzymes able to degrade the ring structure of cholesterol and excess cholesterol must be excreted to avoid a harmful body accumulation. The excretion of cholesterol is mainly achieved in bile via the physiological carriers BA and phospholipids assembled with cholesterol as micelles and vesicles [147]. In addition, the concept of reverse cholesterol transport (RCT) encompasses the pathway transporting excess cholesterol accumulating within peripheral tissues back to the liver for biliary excretion into the feces [148]. FXR stimulates the RCT, as well as the trans-intestinal cholesterol excretion (TICE), the latter representing a significant alternative route to the biliary pathway of RCT. Mechanisms linking FXR to RCT include the changes in the BA pool hydrophobicity and their micellar function with cholesterol [148,149].

A critical role in the relationships between FXR, glucose, fatty acid metabolism and homeostasis seems to be played by the key enzyme pyruvate dehydrogenase kinase 4 (PDK4). In fact, an increased expression of this enzyme has been reported in FXR-null mice, and the inhibition of PDK4 expression alleviated lipid accumulation in hepatocytes both in vivo and in vitro [150].

### 7.2. Glucose Homeostasis and Gluconeogenesis

In both fed and fasted states, the liver is essential to maintain physiological blood glucose levels. Postprandially, exogenous glucose is used in liver to synthesize glycogen and triglycerides. During fasting, glucose is produced after gluconeogenesis and glycogenolysis. FGF15/19 release mediates the effects of FXR on glucose and lipid regulation. FGF15-deficient mice fail to maintain blood concentrations of glucose and normal postprandial amounts of liver glycogen [151]. Activation of the FXR-FGF15/19 axis is involved in glucose metabolism resulting in reduced hepatic gluconeogenesis and glycolysis, and paralleled by increased glycogen synthesis [152]. The mechanisms involve the downstream signaling via SHP-mediated suppression of transcription factors critically involved in gluconeogenesis [153]. Mechanisms might differ between species [154,155] or experimental models, and the role of FXR in glucose metabolism, although evident, is somewhat difficult to interpret. FXR-deficient mice show increased serum glucose levels and impaired glucose and insulin tolerance. By contrast, hepatic gluconeogenesis decreases, and insulin sensitivity improves if FXR is stimulated by feeding the primary BA cholate, treating with the agonist GW4064 or if FXR is overexpressed [142,156]. These results must be interpreted in the context of diet-induced obesity. In aged mice, hepatic loss of FXR and the FXR target SHP improves lipid and glucose homeostasis [157], likely due to an increase in autophagic gene expression (normally repressed by FXR postprandially) [157,158]. 

Additional sets of results originate from the manipulation of intestinal FXR. Mice with selective genetic deletion of intestinal FXR displayed protection against diet-induced diabetes and obesity [75,76]. 

Here, additional complex pathways might play a role. FXR might inhibit the secretion of the intestinal incretin Glucagon-like peptide 1 (GLP-1), as shown by administering the FXR agonist GW4064 [159]. The lack of intestinal FXR would therefore promote the intestinal L-cell release of GLP-1, which improves glycemic control and promotes weight loss. 

In type 2 diabetic mice, the FXR antagonist Mebhydrolin improved glucose homeostasis by suppressing hepatic gluconeogenesis via FXR/miR-22-3p/PI3K/AKT/FoxO1 pathway. An additional effect was the promotion of glycogen synthesis through FXR/miR-22-3p/PI3K/AKT/GSK3β pathway [31].

Fexaramine is the intestine-specific FXR agonist that increases FGF15 signaling, leading to altered BA pool and increasing the level of the secondary tauro-conjugated BA tauro-litocholic acid (TLCA) [160]. TLCA is a strong agonist of the membrane-associated receptor GPBAR-1 in the intestinal (ileum, colon) L-cell, and this step induces the secretion of Glucagon-like peptide 1 (GLP-1). This step appears to explain the improved glucose tolerance and insulin resistance, as well as the stimulation of browning of white adipocytes and energy expenditure i.e., produced heat through uncoupled electron transport in the mitochondria [161,162] by fexaramine in mice [163]. Additional pathways are likely active, since fexaramine treatment in GPBAR-1-deficient mice was also associated with increased energy expenditure [162].

The enterokine FGF15/19 also plays a role in glucose metabolism. FGF19 acts independently from the activity of insulin or the protein kinase Akt. FGF19 acts through a mitogen-activated protein kinase signaling pathway that activates components of the protein translation machinery and stimulates glycogen synthase activity. The processes are tightly regulated to maintain glucose homeostasis. As insulin, FGF15/19 inhibits gluconeogenic gene expression [105]. However, insulin acts through Akt-dependent phosphorylation and subsequent degradation of FOXO1, a transcription factor involved in fasting-mediated induction of gluconeogenic gene expression. Conversely, FGF19 operates promoting dephosphorylation and inactivation of the transcription factor cAMP regulatory element-binding protein (CREB). This effect blunts the expression of peroxisome proliferator-activated receptor gamma coactivator-1alpha (PGC-1α) and other genes involved in hepatic metabolism, such as glucose-6-phosphatase (G6pase). In human and rat hepatocytes and in mouse livers, activation of FXR increased glucose levels via expression of phosphoenolpyruvate carboxykinase (PEPCK) [105,164]. In turn, the overexpression of PGC-1α blocks the inhibitory effect of FGF15/19 on the expression of gluconeogenic gene. In support of this function, mice lacking FGF15 are not able to maintain proper blood concentrations of glucose [151]. 

Notably, mice treated with FGF19 or overexpressing FGF19 have a lower body weight despite elevated food intake. Mice had repression of acetyl-CoA carboxylase 2 (ACC2) and stearoyl-CoA desaturase 1 (SCD1), enhanced energy expenditure and were protected against diet-induced obesity. Decreased ACC2, in turn, decreases mitochondrial Malonyl-CoA levels, and is associated with the upregulation of Carnitine palmitoyltransferase I (CPT1), more availability of fatty acids for oxidation [165]. 

In addition, FGF19 inhibits the synthesis of hepatic fatty acid after suppressing SREBP1c activity [143]. The translational meaning of the above-mentioned studies requires extensive validation [166]. A role for FGF19 at the level of the central nervous system in improving glycemia and peripheral insulin signaling is also possible [167], and points to novel therapeutic targets with antidiabetic drugs. FGF15/19 works in tune with insulin (serum peak of 15 min), as a postprandial regulator of hepatic carbohydrate homeostasis. 

### 7.3. FGF19 and Role in Glycogen Synthesis

Hepatic glycogen synthesis is also under the regulation of FGF15/19. In line with this function and compared to control wild-type mice, *Fgf15* knockout mice become glucose intolerant and store half as much glycogen in the liver. The defect is rescued by administration of FGF19 [151]. Diabetic mice lacking insulin have defective glycogen storage but FGF19 treatment rescues hepatic glycogen concentrations to normal levels. These models suggest that FGF19 activates insulin-independent pathways in regulating glycogen metabolism [151]. To promote the synthesis of glycogen and proteins, FGF19 activates the Ras/ERK pathway, whereas insulin activates the PI3K/Akt pathway [151]. 

Glycogen synthesis in the liver is regulated negatively by glycogen synthase kinase (GSK) 3α and GSK3β. The mechanism is modulated by the effects of BA on phosphorylation and inhibition of glycogen synthase (GS). Phosphorylation also inactivates GSK3 kinases, thus preventing the inhibition of GS and increasing glycogen synthesis. FGF19 acts directly on the liver and induces the phosphorylation of both GSK3α (Ser^21^) and GSK3β (Ser^9^), in parallel with a reduced phosphorylation of Ser^641^ and Ser^645^. In this scenario, GS activity increases, liver glycogen content increases by about 30% without effect on liver cholesterol or triglycerides, as well as insulin or glucagon level. In accord with a direct function of FGF19 on the hepatocyte and glycogen synthesis, it is apparent that fed *Fgf15−/−* mice had >50% less hepatic glycogen and impaired glucose uptake from the circulation, as compared with wild-type animals. FGF19 administration rescued this phenotype completely [151].

Notably, maximal serum insulin levels occur within 1 h of a meal, serum FGF19 levels peak about 3 h after a meal [104] and, in human subjects, liver glycogen levels peak ~4 h after a meal [168]. The conclusive picture which arises from this experimental scenario is that insulin and FGF19 work in a coordinated temporal fashion, and that this synergy facilitates the postprandial storage of nutrients. FGF19, as compared to other anabolic enterokines, namely incretins, GLP-1 and GIP, appears to mimic insulin action independently and without stimulating its release.

### 7.4. FGF19 and Role in Protein Synthesis

In mice, the acute administration of FGF19 increases total protein synthesis by 18% and synthesis of albumin by 40%, while prolonged FGF19 administration increases levels of plasma albumin by 10% [151]. FGF19 stimulates hepatic protein synthesis with the phosphorylation of the ERK1/2, phosphorylation of eIF4B, eIF4E and the ribosomal protein S6 kinase (S6K1), an mTOR- dependent master regulator of muscle cell growth and therefore muscle weight [169]. In primary hepatocytes, the activation of FXR by the agonist INT-747 stimulates amino acids catabolism. In vivo, FXR activation increased ammonium clearance through induction of ureagenesis and glutamine synthesis [170]. In humans, the links between FGF19 and protein synthesis might play a critical role in muscle homeostasis. Lower FGF-19 levels and higher FGF-21 levels have been reported in elderly patients with- as compared to those without sarcopenia, impacting muscle strength [171]. This finding is paralleled by results deriving from animal models, in which antibiotic therapy induced skeletal muscle atrophy secondary to microbial dysbiosis, aberrant BA metabolism and inhibition of the FXR-FGF15 signaling. Of note, in this model, skeletal muscle loss was partly reversed by administration of FGF19 [172].

### 7.5. FGF19 and Role in Energy Expenditure

FGF19 has profound effects on overall metabolism and energy expenditure. The key role of FGF19 in maintaining the physiological homeostasis is testified at various levels and by different models. In the animal models using transgenic mice which express human FGF19 and have decreased fat mass. Mice did not become obese or diabetic when fed a high fat diet. This outcome appears to be the consequence of increased energy expenditure. Likely, FGF19 may increase energy expenditure via increase in brown adipose tissue (BAT) mass. In addition, reduced liver triglyceride levels derive from decreased liver expression of acetyl coenzyme A carboxylase 2 (ACC2) and, in turn, decreased levels of mitochondrially associated malonyl CoA levels and increased activity of carnitine palmitoyl transferase 1 (CPT1), with increased availability of fatty acids for β oxidation [173]. FGF19 increased metabolic rate in mice fed a high fat diet, and this effect was paralleled by reduced body weight and diabetes in leptin-deficient mice [165]. In the central nervous system FGF19 plays an additional role since it improves insulin sensitivity by reducing the activity of hypothalamic agouti-related peptide (AGRP)/neuropeptide Y (NPY) [174]. Thus, FGF15 and FGF19 appear to have several metabolic effects which can be summarized as weight loss, increased insulin sensitivity and thermogenesis. Serum concentrations of cholesterol and triglyceride also decrease with stimulation of FGF15/19. The translational value of such findings, however, requires additional evidence in humans. Postprandially, FGF19 and insulin promote protein and glycogen synthesis and suppresses hepatic gluconeogenesis [89]. At variance with insulin, however, lipogenesis is not stimulated by FGF19, due to different cellular signaling pathways. Taken together, this evidence suggests that FGF15/19 brings beneficial effects on metabolic syndrome and treatment of NASH. Modified FGF19 is beneficial in mouse models of NASH and cholestasis [175]. To harmonize somewhat conflicting results concerning FXR regulation of lipid/glucose metabolism, and possibly metabolic syndrome, it should be noted that FXR knockout mice develop fatty liver, increased serum concentrations of free fatty acids (FFAs) and glucose, and display insulin resistance [156]. In addition, overexpression or activation of hepatic FXR in diabetic db/db and wild-type mice was associated with decreased serum concentrations of FFAs and glucose, and with increased insulin sensitivity [142].

## 8. The Role of FXR in Liver Disease

### 8.1. Inflammation and Fibrosis

Experimental evidence show that FXR can have an anti-inflammatory function in the liver by reducing cholestasis and levels and accumulation of toxic BA [32,176]. The migration and infiltration of monocytes/macrophages is regulated by the chemokine monocyte chemoattractant protein-1 (MCP-1/CCL2) [177]. The synthetic FXR agonist WAY-362450 reduced MCP-1 expression and inflammatory cell infiltration in the liver of mice put on methionine-choline deficient (MCD) diet, which induces NASH. The reduction of hepatic fibrosis by WAY-362450 treatment was paralleled by a reduction in hepatic gene expression of fibrosis markers and was specifically linked to the presence of FXR [32]. The expression of various pro-inflammatory genes is induced by the transcription factor Nuclear factor kappa-light chain enhancer of activated B cell (NF-κB) [178]. Treatment with lipopolysaccharide (LPS) induced in FXR KO mice a strong hepatic inflammation, with massive liver necrosis and marked increase in hepatic cytokine signaling molecules inducible nitric oxide synthase (iNOS), cyclooxygenase-2 (COX-2), and interferon-γ (IFN-γ) [176]. The use of FXR agonists in HepG2 cells and mouse primary hepatocytes suppressed NF-κB mediated inflammation in a FXR-dependent manner [176]. Notably, FXR activation mitigates the development of liver fibrosis due to an increased anti-fibrotic gene expression in hepatic stellate cells (HSCs). The pathway is based on the activation of FXR, induction of SHP, increased expression of peroxisomal proliferator activated receptor γ (PPARγ) with inactivation of HSCs [179,180]. 

In an animal model of cholestasis induced by parenteral nutrition, the administration of the FXR agonist GW4064 prevented hepatic injury and cholestasis. Treated animals showed a normalization of serum BA levels which were associated with increased expression of canalicular bile, of sterol and phospholipid transporters, and with suppression of macrophage recruitment and activation. These effects were secondary to the restoration of hepatic FXR signaling [181].

FGF15/19 can also play a role in inflammation. FGF15 deficiency is associated with the loss of FGF15-mediated suppression of BA synthesis, increased FXR activation, and reduced hepatic fibrosis [182]. In a human HSC cell line, LX2, FGF19 suppresses inflammation through modulating inhibitor of nuclear factor kappa B (IκB) activity without suppression of fibrogenic gene expression [182]. 

### 8.2. Cholestasis

The BA pool is made of amphipathic BA with either protective or toxic effects depending on the tight maintenance of the hydrophilic-hydrophobic balance [183]. Notably, BA structure is responsible for their double signaling function as protective molecules (i.e., proliferation in hepatocytes) or toxic molecules [184]. The process of cholestasis starts with decreased or abolished bile flow. Several conditions may be responsible for cholestasis, but the ultimate step leading to cholestatic liver injury is the intrahepatic accumulation of BA, a situation also defined as BA overload. Chronic BA overload will inevitably cause a progressive BA-induced damage, with the aid of additional toxic components [20,185]. Excess BA retention generates hepatocyte damage, steatosis, fibrosis and even liver tumorigenesis [21,22]. 

BA overload can develop at the hepatic and/or systemic level [186] when trans-hepatocyte BA flow is scarce because of decreased sinusoidal and/or canalicular BA transport [184,187], or in the presence of bile duct obstruction. Both extended (>70%) partial hepatectomy and massive hepatocyte loss [184,188,189,190,191] are a predisposing condition to BA overload [191,192]. Therefore, BA spillover will be evident into the systemic circulation, as confirmed in both animal and human models [19,187,193,194,195,196].

In the model where mice are put on a LCA-enriched diet [197] or to develop impaired pathways of SIRT1 [44], FGF receptors [198,199], SHP [200], FGF15 [201] studies show the expansion of a hydrophobic BA pool which interferes with the liver repairment capacity. Another example is that Cyp2c70-/- mice develop a more human-like hydrophobic BA pool with liver inflammation [202] and altered FXR signaling [203]. This is due to the presence of a more hydrophilic bile in mice, as compared to humans, since the enzyme CYP2c70 (missing in humans) converts CDCA to the more hydrophilic muricholic acid [204]. 

In line with such evidence, BSEP/abcb11-/- mice develop a mild non-progressive cholestasis [197], likely due to an enrichment of the BA pool with hyper-hydroxylated, less hydrophobic, and less cytotoxic BA [205]. In humans, progressive familial intrahepatic cholestasis (PFIC) is an autosomal recessive disease causing 15% of cases of neonatal cholestasis. The PFIC2 form has a mutation in the *ABCB 11* gene encoding BSEP. PFIC patients develop a disrupted secretion of BA from the hepatocytes, progressive hepatic fibrosis, liver cirrhosis and end stage liver disease requiring liver transplantation [206]. Thus, beneficial effects during cholestatic liver injury might derive from an increased hydrophilic profile of the BA pool, either in the animal model [207] and in PFIC children (i.e., by increasing the content of tetrahydroxy BA) [208]. BA overload will cause damage [209] via deranged mitochondrial function [210,211]. This step involves the release of cytochrome c and release of excessive reactive oxygen species (ROS) [212], plasma membrane damage, necrosis, apoptosis and cell death [213]. The indirect damage of excess BA retention involves inflammatory changes associated with cytokine release, neutrophils recruitment and macrophages activation [209,214].

Up to a certain limit, FXR-dependent adaptive responses will try to counteract the cholestatic liver damage. A first preventive mechanism includes the intrahepatic activation of the physiological BA sensor FXR by BA. Both basolateral and canalicular BA transporters and enzymes governing BA synthesis and conjugation are involved [193,215] and contribute to counteracting BA overload in the hepatocytes [216]. FXR-dependent mechanisms can modulate hepatocyte cell cycle progression [192], with a possible regulation of BA homeostasis [114], alcohol-related liver injury [217] and liver regeneration after partial hepatectomy [218,219]. FXR pathways can involve cholangiocyte cell cycle progression [219,220] during BA synthesis suppression [221,222]. The pro-inflammatory effect of BA is mediated by the intracellular assembly of the inflammasome, but FXR exhibits anti-inflammatory effects, because of the interaction with the NLRP3 protein machinery [223]. The modulatory effects of FXR on BA homeostasis and hepatocyte/cholangiocyte cell cycle progression are important to decrease liver BA uptake (inhibition of BA transporters), BA synthesis (suppression of CYP7A1/*Cyp7a1* gene [36,123] and *Cyp8b1* gene [103]), while stimulating BA excretion (activation of BA transporters [124,125]). 

FXR is also expressed by mast cells infiltrating the liver during cholestasis, and promoting hepatic fibrosis. As shown by an animal model, FXR expressed by mast cells has a critical role in hepatic damage and ductular reaction, acting on BA homeostasis through disrupted intestinal and biliary FXR/FGF15 signaling [224].

## 9. FXR as a Therapeutic Target?

The tight interaction between BA-FXR-FGF19 and reflections on metabolism and inflammation in health and disease, has boosted the interest of research on the role of potential therapeutic approaches focusing on FGF19 and FXR stimulation. 

### 9.1. FGF19 and FGF21 Variants

Scarce studies have focused on the role of FGF19 variants devoid of stimulatory effect on FGFR4 (because of potential tumorigenic activity). Mimetic molecules include chimeric molecules FGF19-4, 5 and 6 (mutagenesis in the *N*-terminus and in the heparin binding domains in amino acids 38–42), FGF19v (Conjugation of amino acids 1–20 of FGF21 with amino acids 25–194 of FGF19), M70 (3 amino acid substitutions and 5 amino acid deletions in the *N*-terminus). Such molecules are still able to modulate glucose metabolism but studies are restricted to animal models [225,226,227] with scarce studies in humans [226,228]. The translational value of FGF19 variants is still under evaluation and further evidence is awaited in this field.

FGF21 belong to the FGF19 subfamily of endocrine FGFs and, as FGF19, require the co-receptor β Klotho for binding and signaling through the FGF receptors [229]. As shown in experimental models, FGF21 is a negative regulator of BA synthesis, being able to decrease BA levels in the liver and in the small intestine, with a significant reduction of the BA pool size. These findings are paralleled by decreased colonic and fecal BA, with a concomitant increase in fecal cholesterol and fatty acid excretions [230]. The modulatory effect of FGF21 on BA synthesis seems independent of the FXR/FGF15 pathway [231]. Due to beneficial metabolic effects, FGF21 variants are emerging as promising therapeutic tools in metabolic diseases. The FGF21 variant LY2405319 has been tested in patients with obesity and type 2 diabetes, with beneficial effects on lipid metabolism, body weight, fasting insulin and adiponectin levels, but no significant reduction in fasting glucose levels [228]. Another long-acting FGF21 variant, PF-05231023, induced a body weight loss, an improvement in plasma lipoprotein profile and in adiponectin levels in overweight/obese subjects with type 2 diabetes, without effects on blood glucose. In the treated cohort, however, possible effects on bone formation and resorption were noticed [232]. In obese patients with hypertriglyceridaemia on atorvastatin, with or without type 2 diabetes, the same molecule reduced triglycerides in the absence of weight loss. In the treated group, however, serious adverse effects were noticed, causing the discontinuation of therapy in some participants [233]. Pegbelfermin (BMS-986036), a PEGylated FGF21 analog has been used in obese patients with type 2 diabetes predisposed to fatty liver, resulting in an improvement of the lipid profile, of fibrosis biomarkers and adiponectin levels, in the presence of mild adverse events [234]. Pegbelfermin has been also used in a phase 2a study in obese/overweight subjects and in NASH patients. In this trial, treatment significantly reduced hepatic fat fraction, in the presence of mild side effects (mainly diarrhea and nausea) [235]. Of note, Pegbelfermin promotes a significant reduction from baseline in serum concentrations of DCA and conjugates in patients with NASH and in overweight/obese adults, with a possible modulatory role on the synthesis of secondary BA, also acting on gut microbiome [236]. The extent of total BA decrease recorded in the cited study (about 20–30%) [236] is comparable to that obtained following treatment with FXR agonists [237], with the advantage to be selective for secondary BA [236]. 

### 9.2. FXR Agonists

Clinical trials are on the way using FXR modulators in chronic liver diseases such as primary biliary cholangitis, in cholestasis, nonalcoholic steatohepatitis (NASH), obesity, metabolic syndrome, hypertriglyceridemia, lipodystrophy. Additional trials include bile acid diarrhea, hepatitis B or association with reactivation of latent pro-virus (clinical trials.gov). Table 1 lists the main FXR agonists/modulators explored, at the moment, in clinical trials or experimental studies.

**Table 1 nutrients-14-04950-t001:** FXR agonists/modulators mainly evaluated in clinical or experimental studies.

Obeticholic acid (approved for the treatment of primary biliary cholangitis)	[238,239,240,241]
Tropifexor (LJN452)	[242,243,244,245,246,247,248,249]
Cilofexor (GS-9674)	[250,251]
Vonafexor	[252]
Nidufexor	[253]
GW4064	[254]
MET409	[255]
TC-100	[256]
BMS-986339	[257]
HEC96719	[258]
WAY-450	[259]
WAY-362450	[260]
Px-102	[261]
Px-104	[262]
TERN-101	[263]
EDP-305	[264]
INT-767 (FXR-TGR5 dual agonist)	[25,265,266]

Most solid studies are reporting results with the steroidal molecule obeticholic acid (OCA), the 6α-Ethyl-Chenodeoxycholic Acid (6-ECDCA) and the non-steroidal Tropifexor (LJN452). OCA is modified from CDCA and is about 100 times more potent [238]. 

In male Wistar rats with cholestasis receiving i.v. infusion of LCA to impair bile flow, OCA alone did not induce cholestasis and during co-infusion with LCA, reversed the impairment of bile flow and protected hepatocytes from necrosis [238]. OCA is approved for the treatment of primary biliary cholangitis (PBC) especially in the subgroup of patients who fail to respond to UDCA [239,240,241]. 

Zucker (fa/fa) rats have a loss of function mutation in the hunger hormone leptin receptor [267], and suffer from hyperphagia and hyperleptinaemia resulting in obesity, insulin resistance, diabetes and fatty liver resembling NAFLD [267]. OCA treatment (10 mg/kg/day) over 7 weeks–meaning FXR activation -reversed insulin resistance, prevented body weight gain and fat deposition in the liver, reduced serum levels of triglycerides and aminotransferases, and improved liver histology [267]. Evidence suggests that the activation of FXR by OCA improves hyperglycemia through enhanced insulin secretion and glucose uptake by the liver. OCA increases insulin secretion in mouse β-TC6 cells and human pancreatic islets, while in β-TC6 cells OCA induces AKT (Protein Kinase B)-dependent translocation of glucose transporter 2 (GLUT2). This step increases glucose uptake by these cells [268]. The anti-inflammatory and anti-fibrotic properties of OCA became evident while investigating the NF-κB signaling pathway. In HepG2 cells stimulated with LPS or tumor necrosis factor alpha (TNFα), pretreatment with OCA at 3 uM inhibited the expression of cytokine-inducible enzymes COX-2 and iNOS [176]. OCA inhibited iNOS in LPS-treated primary mouse hepatocytes [176]. 

The agonist activity of OCA on FXR operates also in humans and trials are ongoing [33]. In NASH, OCA showed promising results, with an improvement of liver blood tests, a decreased extent of liver fibrosis and with no worsening of NASH [241]. The efficacy and safety of OCA have been evaluated in patients with type 2 diabetes and NAFLD by a phase IIa study using placebo (n = 23), 25 mg OCA (n = 20), or 50 mg OCA (n = 21) for 6 weeks (ClinicalTrials.gov, Number: NCT00501592). Treatment was well tolerated, and beneficial effects of OCA were observed in both groups with reduced γ-glutamyltransferase (GGT) and alanine aminotransferase (ALT) levels, dose-related weight loss, improved insulin sensitivity, elevated FGF19 serum levels, decreased BA precursor 7a-hydroxy-4-cholesten-3-one (C4) and endogenous BA. Results confirmed the OCA-dependent activation of FXR in human [269].

In the phase IIb trial Farnesoid X Receptor Ligand OCA in NASH Treatment (FLINT), OCA was tested in a multicenter, double-blind, randomized fashion. Non-cirrhotic NASH received 25 mg OCA (n = 141) daily or placebo (n = 142) for 72 weeks. OCA, compared with placebo, improved biochemical and histological features of NASH. NAFLD activity score improved by two points or greater (without worsening of fibrosis) in 45% of OCA patients vs. 21% in the placebo group. Side effects with OCA included pruritus and dyslipidemia [237].

The use of OCA in NASH patients awaits further safety and efficacy data [270]. The phase III trial REGENERATE (by Intercept) was designed to assess the effects of OCA on liver histology and clinical outcomes in 2065 biopsy- confirmed NASH patients. Groups included OCA 10 mg, 25 mg, or placebo for a total duration of six years. The interim analysis was performed by liver biopsy at 18 months in February 2019, and OCA achieved the primary endpoint of improving liver fibrosis without worsening of NASH [271]. 

Results from a REGENERATE 18-Month Interim Analysis in patients with NASH showed improvement of NASH and in health-related quality of life, despite the occurrence of mild pruritus early after the start of OCA therapy [272].

Tropifexor (LJN452), a non-steroidal FXR agonists reduced oxidative stress, steatosis, inflammation and fibrosis in the mouse models of NASH [242]. Tropifexor was safe and well-tolerated following single oral doses ranging from 10 μg to 3 mg. Circulating levels of FGF19 protein increased transiently in a dose-dependent fashion, pointing to a potent on-target FXR agonist activity [273]. Tropifexor evidenced ed a favourable tolerability and pharmacokinetic profile and induced a dose-dependent increase of FGF19 level, with no change in serum lipids in healthy volunteers [243]. In patients with primary bile acid diarrhoea, treatment with tropifexor 60 µg once daily showed an acceptable safety and tolerability profile. Tropifexor treatment decreased 7α-hydroxy-4-cholesten-3-one and bile acid concentration while increased FGF19 level [244]. Ongoing listed studies are focusing on efficacy and tolerability of tropifexor in patients with mild, moderate, severe hepatic impairment or NASH and fibrosis, and primary biliary cholangitis [274]. Recently, in patients with primary biliary cholangitis and inadequate response to UDCA, tropifexor induced a significant improvement in cholestatic markers, in the presence of mild to moderate adverse effects (mainly pruritus) [249].

Thus, further studies are needed with this agonist [245,246,247,248], also with respect to side effects, as compared with OCA. Furthermore, the safety, efficacy, tolerability, pharmacokinetics, pharmacodynamics of novel non-bile acid FXR agonist were intensively examined [275].

Cilofexor (GS-9674), another non-steroidal FXR agonist, evidenced safety and efficacy in non-cirrhotic NASH patients [250]. In detail, a double-blind, placebo-controlled, phase 2 trial was conducted on 140 NASH patients receiving orally cilofexor 100 mg, 30 mg, or placebo once daily for 24 weeks (ClinicalTrials.gov No. NCT02854605.). The study showed that cilofexor was safe and associated with significant attenuation of hepatic steatosis (lipid accumulation), liver enzymes (AST, ALT, and GGT), and bile acids synthesis without any significant alteration of fasting plasma levels of FGF19. In addition, cilofexor treatment was able to reduce different markers of fibrosis and liver stiffness which indicate that cilofexor may have potential antifibrotic effects. In noncirrhotic subjects with large-duct primary sclerosing cholangitis who underwent a 96-week open-label extension of a phase II trial, cilofexor treatment was safe and led to a significant improvement of liver biochemistry and biomarkers of cholestasis and cellular injury [251].

Despite promising results, the side effects and major obstacles of FXR agonist (mainly atherogenic risk, pruritogenic potency) and FGF19 variants (mainly increased appetite, diarrhea and nausea [232,235,276], altered bone homeostasis [232], increased blood pressure and heart rate [233]) still represent a significant matter of concern. Research needs to develop FXR agonists or modulators with beneficial anti-inflammatory effects and minimal metabolic actions. From this point of view, according to available results, the risk-benefit profile of FXR agonists can be modulated by structural optimization of FXR agonist. Harrison et al. [255] evaluated the effect of structurally optimized FXR agonist MET409 in NASH patients. At 12-week post treatment, MET409 reduced fat liver content and bile acid with no significant increase in FGF19 level. The low dose (50 mg) treatment of MET409 showed a 16% pruritus rate and a 9% LDL-C increase but with favorable reduction of fat liver content. The study suggested that the improvements in efficacy/tolerability of FXR agonist treatment could be achieved.

At the molecular level, one oral dose of Px-102, a nonsteroidal FXR agonist, decreased the synthesis of BA in healthy volunteers independently of increases in FGF19 [261] suggesting that that activation of hepatic FXR suppress BA synthesis, independently of FGF19.

The safety and efficacy of FXR agonist Vonafexor for the treatment of chronic hepatits B were also evaluated by a double-blind, placebo-controlled trial. Vonafexor was well tolerated overall with moderate gastrointestinal adverse effects (pruritus occurred in ~60% of subjects with twice-daily treatment compared with 16% in subjects with once-daily treatment). Vonafexor alone or combined with interferon-α2a showed an anti-viral effect by reducing HBV markers [252]. In non-diabetic NAFLD patients, a 4-weeks of treatment with the non-steroidal FXR agonist PX-104 improved liver enzymes and insulin sensitivity with no serious adverse events. Interestingly, PX-104 influenced gut microbiota by reducing *Coriobacteriaceae* abundance and total fecal BAs [262]. TERN-101 (FXR agonist) treatment in healthy volunteer was well-tolerated and showed a reduction in bile acid synthesis (decreases in serum 7α-hydroxy-4-cholesten-3-one) [263].

In an animal model (mice with obstructed BA flow), the novel FXR agonist TC-100 reduced BA pool size in serum and bile, with a shift to a more hydrophilic composition. The changes in BA pool were paralleled by a prevention of intestinal mucosal damage and by a progressive increase in Firmicutes:Bacteroidotes ratio, with increased *Akkermansia muciniphila* abundance [256].

In a recent preclinical study, the BMS-986339, a novel non-bile acid FXR agonist showed potent in vitro and in vivo activation of FXR, with anti-fibrotic efficacy and tissue-selective effects in vivo. The safety of this molecule, however, still requires to be adequately tested in humans [257].

HEC96719, another novel tricyclic FXR agonist, exhibits a potency of FXR activation superior to obeticholic acid, higher FXR selectivity and more favorable tissue distribution in liver and intestine. Although HEC96719 seems a promising tool in NASH treatment, its efficacy and safety profiles need further confirmations [258].

Interestingly, in a rat model of NAFLD/NASH, the administration of dapagliflozin, a sodium-glucose cotransporter-2 inhibitor, alleviated NASH also through a reduced de novo lipogenesis mediated by an upregulation of FXR/SHP and a downregulation of LXRα/SREBP-1c in the liver [277]. These findings parallel a recent observation in patients with T2D and NAFLD, in whom treatment with SGLT2-inhibitors was linked with improvement of liver steatosis and fibrosis markers and circulating pro-inflammatory and redox status [278].

The oral FXR agonist EDP-305 has been proposed for the treatment of NASH. Recent results from a double-blind phase II study in patients with fibrotic NASH showed a significant decrease in ALT levels and liver fat content in 12 weeks, with adverse effects (mainly pruritus) leading to drug discontinuation in about 20% of patients [264]. 

Interesting perspectives derive from experimental studies using FXR-TGR5 dual agonists [25,265,266]. Recently, in a mouse model of NASH, treatment with the FXR-TGR5 dual agonist INT-767 was able to prevent the progression of disease, with beneficial effects on liver mitochondrial function, lipid homeostasis, BA composition (i.e., decreased hydrophobicity index), liver inflammation and gut dysbiosis [25].

## 10. Conclusions and Future Perspectives

Cholesterol is eliminated from human body via biliary secretion and synthesis of primary BA, which are essential for digestion of fat and signaling functions on important receptors. BA homeostasis is a complex scenario which requires several pathways and dynamic events to maintain BA qualitative/quantitative profiles and function in health (Figure 1). 

The effect of BA works in concert with the activation of the nuclear receptor FXR. This step, in turn, controls BA synthesis, excretion, and reabsorption. These pathways minimize over-accumulation of potentially toxic BA in the liver. Activation of FXR also brings several metabolic effects by regulating lipid metabolism, reducing hepatic gluconeogenesis, glycolysis, and increasing glycogen synthesis. In parallel, FXR activation has anti- inflammatory properties during liver injury. The FXR-FGF15/19 axis increases energy expenditure and glycogen synthesis and decreases gluconeogenesis and fatty acid synthesis (Figure 2). FXR and FGF19 have become promising targets for the treatment of NASH. Further studies are required in this important field. 

## Figures and Tables

**Figure 1 nutrients-14-04950-f001:**
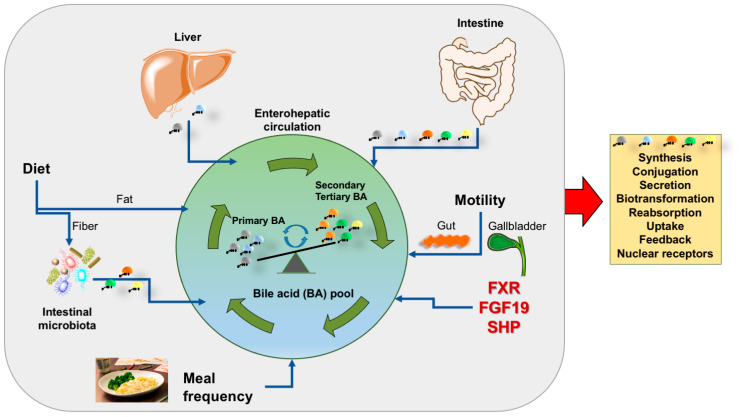
Summary of key factors involved in the composition of the bile acid pool. Several dynamic events are active daily either in the fasting and postprandial period (grey area) and contribute to shape the qualitative/quantitative profile of the bile acid pool, its maintenance and expansion. Such events work in concert with bile acid synthesis and related steps (orange box). Abbreviations: FGF-19, fibroblast growth factor 19; FXR, farnesoid X receptor; SHP, small heterodimer partner; primary bile acids are cholic and chenodeoxycholic acid (grey); secondary bile acids are deoxycholic acid (orange) and lithocholic acid (green); tertiary bile acid is ursodeoxycholic acid (yellow).

**Figure 2 nutrients-14-04950-f002:**
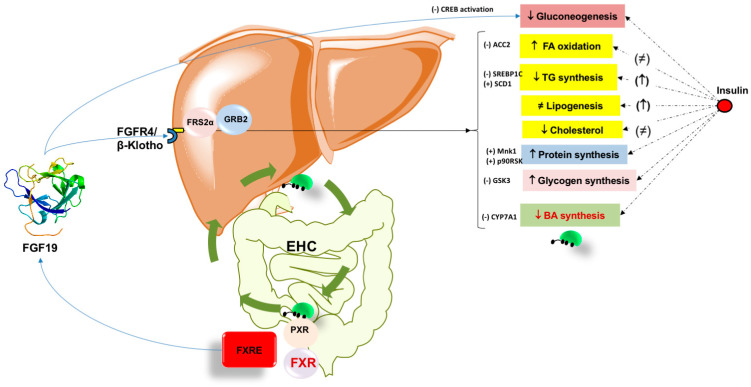
Summary of the metabolic effects of Fibroblast growth factor 19 (FGF19) in the liver. The role of insulin is shown for comparison. Bile acids during the enterohepatic circulation (EHC) activate intestinal FXR promoting FGF19 secretion. Circulating FGF19 signaling requires the presence of β-Klotho [279] which is the fibroblast growth factor receptor 4 (FGFR4) co-receptor required for liver-specific FGF19 actions [112]. The tissue-specific expression pattern of β-klotho and FGFR isoforms determines FGF19 metabolic activity [116]. In the hepatocyte, the signaling events allow the recruitment of cytosolic adaptors such as fibroblast growth factor receptor substrate 2α (FRS2α) and growth factor receptor-bound protein 2 (GRB2). The ultimate metabolic effects of FGF19 are depicted on the right within the boxes, with major regulatory elements listed. Legend; ACC2, acetyl-CoA carboxylase 2; CYP7A1, cholesterol-7α-hydroxylase; CREB cAMP-response element-binding protein; FXR, farnesoid X receptor; FXRE, FXR responsive element; GSK3, glycogen synthase kinase 3; Mnk1, protein kinase; p90 ribosomal S6 kinase; PXR, pregnane X receptor; PXRE, PXR response element; SCD1; stearoyl-CoA desaturase 1; SREBP1C, sterol regulatory element-binding protein 1C. (+) activation; (-) inhibition; ≠, unchanged.

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
