# Peer review of "Recent Advances in the Digestive, Metabolic and Therapeutic Effects of Farnesoid X Receptor and Fibroblast Growth Factor 19: From Cholesterol to Bile Acid Signaling"

_nutrients, 2022, doi:10.3390/nu14234950_

Round 1
Reviewer 1 Report (Previous Reviewer 4)
In this revised version, the authors corrected many errors I pointed out. However, there are still many of typos, grammatical errors, wrong uses of words and even with uncorrected errors pointed earlier. In addition, many statements are very confusing and contain unprofessional remarks. In content wise, the authors’ rigor to cover broad BA and lipid metabolism controlled by FXR would be appreciated. As suggested earlier, this article needs an extensive and careful proofreading again.
1. In line 100, what is the meaning of LRH-1 inhibition? Transcriptional activity or DNA binding? It looks like inhibition of LRH-1 transcriptional activity by SHP. Please rewrite the sentence.
2. In line 103, how does transcription factor FXR regulate enzymatic activities?
3. In line 107, rewrite “coordinates BA detoxification enzyme”. Their gene expression or enzymatic activities (I doubt this)?
4. In line 123, “and to external toxic agents. These”. Change it to “They are also sensitive to external toxic agents, which include”
5. In line 155, “subset” to “in the subset”. “C. leptum increased ----” is very confusing. Please clearly describe the relationship between the microbe and the specific BAs.
6. In line 193, Bacteroides fragilis should be in italic.
7. In line 218, “hortholog”.
8. In line 220, “wich”
9. In line 247, rewrite “The efficacy ---- mitogenic potential”.
10. In 318, “trough”
11. In line 323, what does “FXR-induced increased CYP7A1 expression” mean?
And many more.
In addition, what does each color in BA structures represent for in Fig. 1? Especially, why BAs from liver present the same colors with BAs from intestinal microbiota?
Author Response
Reviewer 1
In this revised version, the authors corrected many errors I pointed out. However, there are still many of typos, grammatical errors, wrong uses of words and even with uncorrected errors pointed earlier. In addition, many statements are very confusing and contain unprofessional remarks. In content wise, the authors’ rigor to cover broad BA and lipid metabolism controlled by FXR would be appreciated. As suggested earlier, this article needs an extensive and careful proofreading again.
We thank the reviewer for the helpful comments. The manuscript has been further revised for type errors and in order to improve clarity. As reported below, the unclear sentences have been rewritten, and an extensive and careful proofreading has been performed by a native English-speaking colleague.
- In line 100, what is the meaning of LRH-1 inhibition? Transcriptional activity or DNA binding? It looks like inhibition of LRH-1 transcriptional activity by SHP. Please rewrite the sentence.
We thank the reviewer for this comment. The sentence has been rewritten as follows:
“This binding leads to increased transcription of the SHP expression. SHP, in turn, inhibits CYP7A1 expression by blocking transactivation of the hepatic activators LRH-1 and hepatic nuclear factor 4 (HNF-4), at the promoter [39]. This pathway ultimately prevents the activation of target genes that participate in BA and fatty acid synthesis”
- In line 103, how does transcription factor FXR regulate enzymatic activities?
We thank the reviewer for this observation. FXR regulates the cited enzymes by transcriptional activity. The sentence has been changed as follows:
“FXR transcriptionally activates the enzymes involved in BA conjugation to glycine or taurine (bile acid-CoA synthetase [BACS] and bile acid-CoA: amino acid N-acetyltransferase [BAT])[43]”
- In line 107, rewrite “coordinates BA detoxification enzyme”. Their gene expression or enzymatic activities (I doubt this)?
The sentence has been changed as follows:
“In addition, FXR activation induces expression of BA detoxification enzymes (i.e., cytosolic sulfotransferase 2A1 (SULT2A1[44]), aldol-keto reductase 1 B7 (AKR1B7[45]), cytochrome P450 3A4/3a11 (CYP3A4/Cyp3a11), and UDP-glycosyltransferase 2B4 (UTG2B4)) [46].”
- In line 123, “and to external toxic agents. These”. Change it to “They are also sensitive to external toxic agents, which include”
The sentence has been changed as suggested.
- In line 155, “subset” to “in the subset”. “C. leptum increased ----” is very confusing. Please clearly describe the relationship between the microbe and the specific BAs.
We agree with the reviewer that the sentence was very confusing. The whole paragraph has been now changed as follows:
“Compared to healthy controls, higher levels of total faecal BA, primary CA and CDCA, and higher BA synthesis have been reported in patients with NASH, who also showed a higher ratio of primary to secondary BA, but a similar ratio of conjugated to unconjugated BA. Patients with NASH were also characterized by a decreased count of Bacteroidetes and Clostridium leptum. The count of C. leptum increased with fecal unconjugated LCA and decreased with unconjugated CA and CDCA. Taken together, these findings point to a link between NAFLD, dysbiosis and altered BA homeostasis, which puts patients at risk of further hepatic injury [71].”
- In line 193, Bacteroides fragilis should be in italic.
The text has been changed as suggested.
- In line 218, “hortholog”.
Sorry for this type error. The word has been corrected and checked throughout the text.
- In line 220, “wich”
Sorry for this type error. The word has been corrected.
- In line 247, rewrite “The efficacy ---- mitogenic potential”.
We thank the reviewer for this suggestion. The sentence has been rewritten as follows:
“Therapies involving FGF19 seem promising. However, the translational value of these attractive therapeutic tools should be carefully verified in terms of possible hepatic tu-morigenesis, as a consequence of their mitogenic potential [107,108].”
- In 318, “trough”
Sorry for this type error. The word has been corrected.
- In line 323, what does “FXR-induced increased CYP7A1 expression” mean?
We agree with the reviewer that the sentence was not clear. The text has been now changed as follows:
“Further studies in FXR-deficient and CYP7A1-deficient mice confirmed the beneficial effects on hepatic inflammation of the increased CYP7A1 expression and BA synthesis secondary to activation of FXR [27].”
And many more.
We further revised the manuscript for type errors and in order to improve clarity.
In addition, what does each color in BA structures represent for in Fig. 1? Especially, why BAs from liver present the same colors with BAs from intestinal microbiota?
Thank you for pointing this aspect. We now provided different colours for primary, secondary, and tertiary BA, according to specific sites (liver, microbiota, intestine, and processes). We changed the figure legend accordingly:
“Figure 1. Summary of key factors involved in the composition of the bile acid pool. Several dy-namic events are active daily either in the fasting and postprandial period (grey area) and con-tribute to shape the qualitative/quantitative profile of the bile acid pool, its maintenance and ex-pansion. Such events work in concert with bile acid synthesis and related steps (orange box). Ab-breviations: FGF-19, fibroblast growth factor 19; FXR, farnesoid X receptor; SHP, small hetero-dimer partner; primary bile acids are cholic and chenodeoxycholic acid (grey); secondary bile acids are deoxycholic acid (orange) and lithocholic acid (green); tertiary bile acid is ursodeoxy-cholic acid (yellow).”
Reviewer 2 Report (Previous Reviewer 3)
Careful editing is needed to improve clarity and readability.
teria or inhibiting other bile sensitive bacteria. Gut Eubacterium lentum, Ruminococcus 139
Add a hyphen between “bile” and “sensitive”!
profoundly affected the microbiota, metabo-172 lome profile mucosal biomarkers of cancer risk when African Americans were fed a 173
There should be a comma between profile and mucosal.
causes SIBO can be reversed by administration of BA [77,78]. A study in the cholestatic 192
Spell out SIBO!
Both extended (>70%) partial 566 hepatectomy or massive hepatocyte loss [184,188-191] are a predisposing condition to BA 567
Replace “or” with “and”!
This step involves the release of cytochrome c aGnd release of exces-588
Replace “aGnd” with “and”!
FXR-dependent mechanisms can modulate hepatocyte cell cycle progression [192] trying 598 to govern for example BA homeostasis [114], alcohol-related liver injury [217] and liver 599 regeneration after partial hepatectomy [218,219].
What does “trying” mean between cell cycle progression and BA homeostasis?
The following sections require better formatting!
mary biliary cholangitis 751 (https://clinicaltrials.gov/ct2/results?cond=&term=tropifexor&cntry=&state=&city=&dist=752
sively examined 759 (https://clinicaltrials.gov/ct2/results?cond=&term=FXR+agonist&cntry=&state=&city=&di760 st=&Search=Search) . 761
Author Response
Point-by-point answers to reviewers
Reviewer 2
Careful editing is needed to improve clarity and readability.
We thank the reviewer for his/her suggestions. The paper has been carefully checked to improve clarity.
- teria or inhibiting other bile sensitive bacteria. Gut Eubacterium lentum, Ruminococcus 139
Add a hyphen between “bile” and “sensitive”!
The text has been changed as suggested
- profoundly affected the microbiota, metabo-172 lome profile mucosal biomarkers of cancer risk when African Americans were fed a 173
There should be a comma between profile and mucosal.
The text has been changed as suggested
- causes SIBO can be reversed by administration of BA [77,78]. A study in the cholestatic 192
Spell out SIBO!
The text has been changed as follows:
“In rodents, the biliary obstruction that causes small intestinal bacterial overgrowth (SIBO) can be reversed by administration of BA”
- Both extended (>70%) partial 566 hepatectomy or massive hepatocyte loss [184,188-191] are a predisposing condition to BA 567
Replace “or” with “and”!
The text has been changed as suggested
- This step involves the release of cytochrome c aGnd release of exces-588
Replace “aGnd” with “and”!
The text has been changed as suggested
- FXR-dependent mechanisms can modulate hepatocyte cell cycle progression [192] trying 598 to govern for example BA homeostasis [114], alcohol-related liver injury [217] and liver 599 regeneration after partial hepatectomy [218,219].
What does “trying” mean between cell cycle progression and BA homeostasis?
The text has been changed as follows:
“FXR-dependent mechanisms can modulate hepatocyte cell cycle progression [192], with a possible regulation of BA homeostasis [114],…”
- The following sections require better formatting!
mary biliary cholangitis 751 (https://clinicaltrials.gov/ct2/results?cond=&term=tropifexor&cntry=&state=&city=&dist=752
The link has been now inserted as in-text citation:
“Ongoing listed studies are focusing on efficacy and tolerability of tropifexor in patients with mild, moderate, severe hepatic impairment or NASH and fibrosis, and primary biliary cholangitis [274]”.
- sively examined 759 (https://clinicaltrials.gov/ct2/results?cond=&term=FXR+agonist&cntry=&state=&city=&di760 st=&Search=Search) . 761
The link has been now inserted as in-text citation:
“….tolerability, pharmacokinetics, pharmacodynamics of novel non-bile acid FXR agonist were intensively examined [275].

Reviewer 3 Report (Previous Reviewer 2)
The authors have adequately addressed all the issues that the reviewers raised for their original submission.
Author Response
We sincerely thank the reviewer for the helpful comments and suggestions.

Round 2
Reviewer 1 Report (Previous Reviewer 4)
The authors were expected to do proofreading extensively. There are still many errors and unprofessional statements which are literally difficult to understand.
The followings are those errors I was able to spot.
However, still many statements are awkward and confusing.
Please proofread the manuscript.
More errors and awkward statements I spotted
1. In line 498, “FGF19 increase energy ---- (BAT)”. Elaborate the sentence. Energy expenditure or production? BAT function or mass?
2. In line 499, ACC2 is not the rate-limiting enzyme for FA entry to mitochondria. The rate limiting enzyme is CPT1A. Please read the ref article carefully and thoroughly. ACC2 may block CPT1A activity.
3. In line 514, what is FXR gelation?
4. In line 588, “aGnd”.
5. In line 604, what specific steps do authors talk about?
6. Line 608. “can fail to counteract and prevent”. Hard to understand?
7. Line 639. Induced a body weight loss, ------ in OW/obese subjects with T2D.
8. Line 811. HEC96719, (place comma).
9. Line 814. also in this case, (place comma). Or simply its efficacy.
Author Response
Point-by-point answers to reviewers
Reviewer 1
The authors were expected to do proofreading extensively. There are still many errors and unprofessional statements which are literally difficult to understand.
The followings are those errors I was able to spot.
However, still many statements are awkward and confusing.
Please proofread the manuscript.
More errors and awkward statements I spotted
We thank the reviewer for his/her suggestions. The paper has been further checked to improve clarity.
- In line 498, “FGF19 increase energy ---- (BAT)”. Elaborate the sentence. Energy expenditure or production? BAT function or mass?
We thank the reviewer for this comment. the sentence has been changed as follows:
“FGF19 may increase energy expenditure via increase in brown adipose tissue (BAT) mass”
- In line 499, ACC2 is not the rate-limiting enzyme for FA entry to mitochondria. The rate limiting enzyme is CPT1A. Please read the ref article carefully and thoroughly. ACC2 may block CPT1A activity.
We thank the reviewer for this observation. The sentence has been revised as follows:
“reduced liver triglyceride levels derive from decreased liver expression of acetyl coenzyme A carboxylase 2 (ACC2) and, in turn, decreased levels of mitochondrially associated malonyl CoA levels and increased activity of carnitine palmitoyl transferase 1 (CPT1), with increased availability of fatty acids for β oxidation [173].”
- In line 514, what is FXR gelation?
Sorry for the type error. The word has been changed in “regulation”
- In line 588, “aGnd”.
Sorry for the type error. The word has been changed in “and”
- In line 604, what specific steps do authors talk about?
We agree that the sentence was not clear. The text has been changed as follows:
“The modulatory effects of FXR on BA homeostasis and hepatocyte/cholangiocyte cell cycle progression are important to decrease liver BA uptake (inhibition of BA transporters), BA synthesis (suppression of CYP7A1/Cyp7a1 gene [36,123] and Cyp8b1 gene [103]), while stimulating BA excretion (activation of BA transporters [124,125]).”
- Line 608. “can fail to counteract and prevent”. Hard to understand?
The concept was redundant. This sentence has been now deleted from the text
- Line 639. Induced a body weight loss, ------ in OW/obese subjects with T2D.
We thank the reviewer for this suggestion. The sentence has been changed as follows:
“Another long-acting FGF21 variant, PF-05231023, induced a body weight loss, an improvement in plasma lipoprotein profile and in adiponectin levels in overweight/obese subjects with type 2 diabetes, without effects on blood glucose.”
- Line 811. HEC96719, (place comma).
The text has been changed as suggested.
- Line 814. also in this case, (place comma). Or simply its efficacy.
The sentence has been changed as follows:
“Although HEC96719 seems a promising tool in NASH treatment, its efficacy and safety profiles need further confirmations”
This manuscript is a resubmission of an earlier submission. The following is a list of the peer review reports and author responses from that submission.
Round 1
Reviewer 1 Report
This review by Di Ciaula et al. nicely summarizes current literature on bile acid metabolism, signaling, and disease treatment. Although the authors provide much and very comprehensive information on bile acid synthesis and metabolism, as well as FGF15/19 and FXR, this review is partly very similar to the previous one published in 2017 by Di Ciaula et al. as well as the review published in 2020 by Portincasa et al..
The Reviewer has to reject the present manuscript but suggests that the authors shorten the review and reduce it to describing novel findings (i.e. what has not already been summarized in previous reviews).
Major concerns:
Some parts of the manuscript are highly similar to the Review from 2017, including Figure1 (old Figure 1), Figure 2B (old Figure 3) and Figure 3A (old Figure 4). The authors only applied very minor modifications compared to the indicated sources, while for Figure 1 no respective reference is indicated.
Other Figures, including Figure 2B, 3B, and 6 are – although the authors cited the source were pictures were adapted from – more or less completely identical to the one published by Portincasa et al. 2020. Respective Figures from 2020 are Figures 1, 2, and 5.
Additionally, all Tables used in this review were adapted from the previous publication from Portincasa et al.. Despite correct references for each Table and most Figures (except Figure 1), this manuscript represents a combination of old reviews rather than a summary of new findings.
When writing a review, it is of utmost importance that the authors cite primary literature and research articles, not summarizing old reviews or whole book chapters. Skimming through the first 15 references, the Reviewer had to realize that except 3 sources, all literature are reviews.
Author Response
|
Reviewer 1 |
|
This review by Di Ciaula et al. nicely summarizes current literature on bile acid metabolism, signaling, and disease treatment. Although the authors provide much and very comprehensive information on bile acid synthesis and metabolism, as well as FGF15/19 and FXR, this review is partly very similar to the previous one published in 2017 by Di Ciaula et al. as well as the review published in 2020 by Portincasa et al.. The Reviewer has to reject the present manuscript but suggests that the authors shorten the review and reduce it to describing novel findings (i.e., what has not already been summarized in previous reviews). |
|
ANSWER: We thank the reviewer for the appreciation and for the helpful suggestions. We originally thought that the readers of Nutrients could be a different target audience compared with readers in the Hepato-gastrointestinal field. Thus, we planned an exhaustive review article with several didactic and explanatory figures, freely available from previous papers from our group or totally drawn by us (P.P.). However, we take the point of this reviewer and for this the paper has been shortened by at least 40%, previous old figures 1-7 have been eliminated (there are only 2 figures left, drawn by one of the authors), and the 2 tables eliminated. Recent references on most relevant aspects have been included. The review has been re-assembled according to the topic of the present manuscript. Here, we also report and discuss the most relevant novel findings. |
|
Major concerns: Some parts of the manuscript are highly similar to the Review from 2017, including Figure1 (old Figure 1), Figure 2B (old Figure 3) and Figure 3A (old Figure 4). The authors only applied very minor modifications compared to the indicated sources, while for Figure 1 no respective reference is indicated. Other Figures, including Figure 2B, 3B, and 6 are – although the authors cited the source were pictures were adapted from – more or less completely identical to the one published by Portincasa et al. 2020. Respective Figures from 2020 are Figures 1, 2, and 5. Additionally, all Tables used in this review were adapted from the previous publication from Portincasa et al.. Despite correct references for each Table and most Figures (except Figure 1), this manuscript represents a combination of old reviews rather than a summary of new findings. When writing a review, it is of utmost importance that the authors cite primary literature and research articles, not summarizing old reviews or whole book chapters. Skimming through the first 15 references, the Reviewer had to realize that except 3 sources, all literature are reviews. |
|
ANSWER: We thank the reviewer for these observations which we took into consideration on a point-by-point basis and incorporating also the first remark of this reviewer. In particular, the cited figures/tables have been now removed from the manuscript. Only two figures totally new have been included and relate to the section “Conclusions and future perspectives”. Figure 1. Summary of key factors involved in the composition of the bile acid pool. Several dynamic events are active daily either in the fasting and postprandial period (grey area) and contribute to shape the qualitative/quantitative profile of the bile acid pool, its maintenance and expansion. Such events work in concert with bile acid synthesis and related steps (orange box). Abbreviations: FGF-19, fibroblast growth factor 19; FXR, farnesoid X receptor; SHP, small heterodimer partner; (I), primary bile acids; (II), secondary bile acids. Figure 2. Summary of the metabolic effects of Fibroblast growth factor 19 (FGF19) in the liver. The role of insulin is shown for comparison. Bile acids during the enterohepatic circulation (EHC) activate intestinal FXR promoting FGF19 secretion. Circulating FGF19 signalling requires the presence of β Klotho [300] which is the fibroblast growth factor receptor 4 (FGFR4) co receptor required for liver-specific FGF19 actions [119]. The tissue-specific expression pattern of β klotho and FGFR isoforms determines FGF19 metabolic activity [123]. In the hepatocyte, the signalling events allow the recruitment of cytosolic adaptors such as fibroblast growth factor receptor sub-strate 2α (FRS2α) and growth factor receptor-bound protein 2 (GRB2). The ultimate metabolic effects of FGF19 are depicted on the right within the boxes, with major regulatory elements listed. Legend; ACC2, acetyl-CoA carboxylase 2; CYP7A1, cholesterol‑7α‑hydroxylase; CREB cAMP-response element-binding protein; FXR, farnesoid X receptor; FXRE, FXR responsive ele-ment; GSK3, glycogen synthase kinase 3; Mnk1, protein kinase; p90 ribosomal S6 kinase; PXR, pregnane X receptor; PXRE, PXR response element; SCD1; stearoyl-CoA desaturase 1; SREBP1C, sterol regulatory element-binding protein 1C. (+) activation; (-) inhibition; ≠, unchanged. In addition, the text has been extensively revised focusing on the main topic of the present review article. The most relevant novel findings (i.e., research articles published in the years 2020-2022) have been now cited and discussed in the different sections of the paper. All changes appear in red color. |
Reviewer 2 Report
In this manuscript, Di Ciaula et al summarized the recent progress in both of basic biology research and clinical studies for FXR and FGF19 signaling in the liver and metabolic disease. The authors provided a broad, with the great depth in the meantime, of our current understanding of FXR/FGF19 pathway in regulating of BA hemostasis, lipid and glucose metabolism, energy expenditure and its role in liver diseases. In general, the manuscript is well written, and it would be a great plus for the journal if it is accepted for publication. Some issues need to be addressed are as below:
1. Some contents and the figures could be more concise to avoid the redundant information in different sections of the manuscript. For example, the BA reabsorption route (enterohepatic circulation) and FXR signaling on BA synthesis inhibition etc.
2. Need to summarize and provide some insight about the side effects and the major obstacles of FXR agonist including FGF19 variants in clinical trials for NAFLD/NASH treatment.
Minor issue:
1. Table 1 need to be reformatted so it can be readable.
2. Need to rephrase the sentence “increased the synthesis of the FXR antagonist TβMCA increases (Page 15, line14).
Author Response
|
Reviewer 2 |
|
In this manuscript, Di Ciaula et al summarized the recent progress in both of basic biology research and clinical studies for FXR and FGF19 signaling in the liver and metabolic disease. The authors provided a broad, with the great depth in the meantime, of our current understanding of FXR/FGF19 pathway in regulating of BA hemostasis, lipid and glucose metabolism, energy expenditure and its role in liver diseases. In general, the manuscript is well written, and it would be a great plus for the journal if it is accepted for publication. Some issues need to be addressed are as below: |
|
ANSWER: We sincerely thank this reviewer for appreciating the work and the informative profile of our review. As discussed with Reviewer 1, we originally thought that the readers of Nutrients could be a different target audience compared with readers in the Hepato-gastrointestinal field. In line with this thought, we planned an exhaustive review article with several didactic and explanatory figures, freely available from previous papers from our group or totally drawn by us (P.P.). However, the suggestions from this reviewer allow us to improve the message and the paper itself. In particular, the paper has been shortened by at least 40%, previous old figures 1-7 have been eliminated (there are only 2 figures left, drawn by one of the authors), and the 2 tables eliminated. Recent references on most relevant aspects have been included. |
|
1. Some contents and the figures could be more concise to avoid the redundant information in different sections of the manuscript. For example, the BA reabsorption route (enterohepatic circulation) and FXR signaling on BA synthesis inhibition etc. |
|
ANSWER: Thank you for suggestions which re in line with those from reviewer 1. The first 7 figures in R0 have been eliminated and now only 2 figures (never appearing in our previous publications) have been kept in the section “Conclusions and future perspectives”. In addition, and as suggested, we discuss only non-redundant information in different sections of the manuscript |
.
|
2. Need to summarize and provide some insight about the side effects and the major obstacles of FXR agonist including FGF19 variants in clinical trials for NAFLD/NASH treatment. |
|
ANSWER: We thank the reviewer for this suggestion. The following sentence has been added in the text (page 16 para 9.2 FXR agonists): “Despite promising results, the side effects and major obstacles of FXR agonist (mainly atherogenic risk, pruritogenic potency) and FGF19 variants (mainly increased appetite, diarrhea and nausea [268,271,294], altered bone homeostasis [268], increased blood pressure and heart rate[269]) still represent a significant matter of concern. Future studies should focus on the development of FXR agonists or modulators having beneficial an-ti-inflammatory effects with poor metabolic actions. From this point of view, according to available results, the risk-benefit profile of FXR agonists can be modulated by structural optimization of FXR agonist.” |
|
Minor issue: 1. Table 1 need to be reformatted so it can be readable. |
|
ANSWER: Table 1 has been now deleted from the paper |
|
2. Need to rephrase the sentence “increased the synthesis of the FXR antagonist TβMCA increases (Page 15, line14). |
|
ANSWER: The sentence has been changed as follows: “In an animal model of NAFLD induced by high-fat diet, the antibiotic treatment de-creased BSH-encoding Lactobacillus, increased the synthesis of the FXR antagonist TβMCA, and improved insulin resistance and liver steatosis[86]” |
Reviewer 3 Report
The review presents an interesting update on FXR-FGF19 in BA signaling in multiple tissues and diseases with more than 300 citations. Despite its comprehensiveness, the review is diffusive, extremely poorly edited, and littered with typographical and grammatical errors. The text consist of long and winding paragraphs that provide just brief mentioning of main conclusions without assessment of the significance and limitations of cited studies. What is even more annoying is that a lot of sentences were simply cobbled together without logical connection. In the end, no clear picture emerges to give the readers an update on the current research and future directions. Biosynthesis, secretion, absorption, and circulation of BA have taken too much space and could have been easily converted into a stand-alone review. Most illustrations contain many labels and bewildering lines, making it very difficult to figure out how all the molecules are involved and how pathways are regulated and coordinated in different tissues.
Author Response
|
.Reviewer 3 |
|
The review presents an interesting update on FXR-FGF19 in BA signaling in multiple tissues and diseases with more than 300 citations. Despite its comprehensiveness, the review is diffusive, extremely poorly edited, and littered with typographical and grammatical errors. The text consist of long and winding paragraphs that provide just brief mentioning of main conclusions without assessment of the significance and limitations of cited studies. What is even more annoying is that a lot of sentences were simply cobbled together without logical connection. In the end, no clear picture emerges to give the readers an update on the current research and future directions. Biosynthesis, secretion, absorption, and circulation of BA have taken too much space and could have been easily converted into a stand-alone review. Most illustrations contain many labels and bewildering lines, making it very difficult to figure out how all the molecules are involved and how pathways are regulated and coordinated in different tissues.
|
|
ANSWER: We thank the reviewer for these comments. The paper has been now extensively revised, re-edited and shortened by at least 40%. Previous old figures 1-7 have been eliminated, as also the 2 tables. Recent references on the most relevant aspects have been included and commented in the revised version of the manuscript. The review has been reassembled according to the topic. Language revised thoroughly. All changes appear now in red colour throughout the text. |
Reviewer 4 Report
The current review covered physiological roles of FXR in metabolic pathways especially focusing on BA homeostasis. The authors also cited up-to-date literatures to bring up significant advances of therapeutic efficacies of FXR activation by various synthetic FXR agonists in BA associated metabolic disorders. The authors summarized clear and data supported FXR roles in various metabolic pathways rigorously. There are some redundant information in various pages, which fortunately help readers understand the content more clearly. Though the review coverage is very good and wide, spotted many errors (grammatical and spelling) should be corrected before publication.
In addition, I have a few suggestions that are listed below:
1. Mechanistic details of intestinal FXR antagonism against NASH development are missing in page 15. In addition, human FXR BA antagonists were also reported (Nat Med, 2018, 24(12), p1919).
2. From the provided info in the article, lowering hepatic BA synthesis would be beneficial for dealing with various metabolic syndrome. However, beneficial effect of cholesterol catabolism (BA synthesis) should be also commented. Hepatic accumulation of cholesterol has been considered a critical culprit for NASH development. Especially additional insights on contradictory results from mouse studies (Cyp7a overexpression vs Cyp7a deletion, Hepatology 2010, 52, p679 and JLR 2016, 57, p1144 from the same research group) would be valuable.
3. In Fig, 2, small BA structures may be shown with green or yellow circles as primary or secondary BAs. However, in Fig. 3B, Fig. 8, and Fig. 9, the color doesn’t look the same as described in Fig. 2. Please, indicate them correctly in each figure legend, especially for the colors shown in figure 8.
There are many incorrect uses (missing or addition) of “comma” and “and”, which makes understanding sentences a little difficult. Only a few examples of errors other than “comma” or “and” are listed below. The article including figure legends needs extensive proofreading.
1. In page 18, glucagon-like peptide 1 should be GLP-1 not GPBAR-1.
2. Small heterodimer partner should be SHP not SHP-1 in order not to be confused with Src-homology 2 domain containing PTPs, SHP-1 and SHP-2. The authors used both SHP and SHP-1 in the manuscript.
Author Response
|
Reviewer 4 |
|
The current review covered physiological roles of FXR in metabolic pathways especially focusing on BA homeostasis. The authors also cited up-to-date literatures to bring up significant advances of therapeutic efficacies of FXR activation by various synthetic FXR agonists in BA associated metabolic disorders. The authors summarized clear and data supported FXR roles in various metabolic pathways rigorously. There are some redundant information in various pages, which fortunately help readers understand the content more clearly. Though the review coverage is very good and wide, spotted many errors (grammatical and spelling) should be corrected before publication.
ANSWER: we thank the reviewer for his/her appreciation and for providing helpful comments. The paper has been extensively reviewed to avoid redundant information. At the same time, the text has been re-edited and shortened by at least 40%, to improve clarity and to better focus on the main topic of the review. Grammatical and spelling errors have been carefully checked throughout the manuscript.
In addition, I have a few suggestions that are listed below:
1. Mechanistic details of intestinal FXR antagonism against NASH development are missing in page 15. In addition, human FXR BA antagonists were also reported (Nat Med, 2018, 24(12), p1919).
ANSWER: we thank the reviewer for this suggestion. We apologize for missing this important information which helps clarify an important function of FXR and BA. The following text has been now added in the manuscript, concerning intestinal FXR antagonism (Paragraph 4):
“In an animal model of NAFLD induced by high-fat diet, the antibiotic treatment decreased BSH-encoding Lactobacillus, increased the synthesis of the FXR antagonist TβMCA, and improved insulin resistance and liver steatosis [86]. In humans with newly diagnosed type 2 diabetes, naïve treatment with metformin modified gut microbiota, decreasing Bacteroides fragilis and increasing the BA glycoursodeoxycholic acid (GUDCA) in the gut. These changes were paralleled by an inhibition of FXR signaling, pointing to GUDCA as an intestinal FXR antagonist able to improve metabolic dysfunction [87]”.
2. From the provided info in the article, lowering hepatic BA synthesis would be beneficial for dealing with various metabolic syndrome. However, beneficial effect of cholesterol catabolism (BA synthesis) should be also commented. Hepatic accumulation of cholesterol has been considered a critical culprit for NASH development. Especially additional insights on contradictory results from mouse studies (Cyp7a overexpression vs Cyp7a deletion, Hepatology 2010, 52, p679 and JLR 2016, 57, p1144 from the same research group) would be valuable.
ANSWER: we sincerely thank the reviewer for this comment. The text on BA physiology has been shortened and re-edited in the revised version of the manuscript, underlying the links between BA synthesis/cholesterol catabolism/NAFLD in the new section 7.1 (“Lipid homeostasis”). The following text has been added in the manuscript, also discussing the suggested refs:
“The pathways connecting cholesterol to BA synthesis are essential and contribute to the prevention of pathological amounts of cholesterol in the body [1,7], as also to metabolic homeostasis. BA synthesis starts from cholesterol in the liver, and this step represents the main catabolic pathway of cholesterol metabolism in humans [7]. These pathways act through the “classic” neutral pathway (cholesterol 7α-hydroxylase, CYP7A1), contributing to about 75% of the total BA pool, and the “alternative” acidic pathway (sterol 27-hydroxylase, CYP27A1) contributing to about 25% of the total BA pool [7,157]. In this context, FXR has a relevant role in atherosclerotic risk factors, and strongly modulates the homeostasis of cholesterol due to the ability to inhibit CYP7A1. Of note, increased serum levels of primary and secondary BA have been reported in patients with NAFLD, as compared with healthy controls [48]. This finding should suggest a role for the elevated BA production in the pathogenesis of NAFLD. This link, however, seems secondary to the increased proportion, in the BA pool, of the FXR antagonistic DCA, and to decreased levels of the FXR agonistic CDCA [48]. On the other hand, the cholesterol catabolism trough BA synthesis has beneficial effects on metabolic homeostasis, and hepatic accumulation of cholesterol is a critical culprit for the development of NAFLD/NASH[158]. Transgenic mice overexpressing CYP7A1 in the liver are resistant to high-fat diet-induced obesity, fatty liver and insulin resistance, mainly through increased hepatic cholesterol catabolism and increased BA pool [159]. Further studies in FXR-deficient and CYP7A1-deficient mice confirmed the beneficial effects of FXR-induced increased CYP7A1 expression and BA synthesis on hepatic inflammation[160].”
3. In Fig, 2, small BA structures may be shown with green or yellow circles as primary or secondary BAs. However, in Fig. 3B, Fig. 8, and Fig. 9, the color doesn’t look the same as described in Fig. 2. Please, indicate them correctly in each figure legend, especially for the colors shown in figure 8.
ANSWER: the cited figures have been now deleted from the manuscript as suggested by other reviewers. We believe that too many figures were repetitive of some already established concepts. I hope this reviewer agrees with the drastic reduction from 9 to 2 figures (the last two of the previous version).
There are many incorrect uses (missing or addition) of “comma” and “and”, which makes understanding sentences a little difficult. Only a few examples of errors other than “comma” or “and” are listed below. The article including figure legends needs extensive proofreading.
ANSWER: the paper has been extensively revised. Grammatical and spelling errors have been carefully checked throughout the manuscript. Thank you for pointing this out.
1. In page 18, glucagon-like peptide 1 should be GLP-1 not GPBAR-1.
ANSWER: thanks for this comment. the error has been now corrected.
2. Small heterodimer partner should be SHP not SHP-1 in order not to be confused with Src-homology 2 domain containing PTPs, SHP-1 and SHP-2. The authors used both SHP and SHP-1 in the manuscript.
ANSWER: thanks for this comment referring to a typing error. The small heterodimer partner has been now indicated as SHP throughout the manuscript |
Round 2
Reviewer 1 Report
The manuscript has much improved compared to the initial version and many suggestions of the Reviewers have been addressed or discussed. However, there is still high need for improvement, especially in terms of grammar and the use of some very confusing sentences without much meaning. The abstract contains too much information (it is not necessary to describe all the pathways) and can be shortened. Additionally, the authors have not addressed all of the issues raised, particularly with regard to the correct citation of primary sources and the inclusion of more recent literature (please see comments below).
Major concerns:
1. The introduction is only citing 3 papers, all published by the corresponding author, ignoring other literature – in particular, no primary sources are cited. More than 70% of the introduction only summarizes summary of a previous review/book chapter, without any primary literature cited.
2. The following statement on biliary lipids is not correct: “In bile, BA are the major lipid component, followed by phospholipids and almost totally unesterified cholesterol. Overall, the three lipids account for about 99% of total lipids by weight. Bilirubin is another lipid in bile and represents less than 1% of biliary lipids.” The three main lipids of bile are bile salts, phospholipids and cholesterol, which is nonesterified. Free cholesterol accounts for 97% of all sterols in bile, the rest are cholesterol precursors and dietary phytosterols. Additionally, “Almost totally unesterified cholesterol” sounds misleading, as it indicates the existence of partially esterified cholesterol as well, which is incorrect. Please be aware of that. Furthermore, bilirubin is not a lipid species itself and it is present in bile mainly as bilirubin-conjugates.
3. There is no reference indicated for the last paragraph of introduction (describing consequences of disrupted BA homeostasis).
4. Redundant information: “BA lost in feces daily account for 5% and this amount equals the daily synthesis in the liver (0.2-0.6 g/daily). The enterohepatic circulation of BA consists of continuous recirculation of BA (4-12 times daily) from the intestine to the liver and back to the intestine, depending on meal type and frequency.” (Chapter 2) vs. already described in introduction: “By this dynamic mechanism, BA undergo continuous enterohepatic circulation with 4-12 cycles/day. The fecal loss is minimal at every cycle (5%) and must be compensated by the de novo synthesis of primary BA in the liver (~200-600 mg/daily).”
5. Checking literature in the revised manuscript revealed redundant citations. For example, why does the following sentence need two references!? “The cycling of BA between the liver and the intestine is referred to as the enterohepatic circulation, while the total amount of BA in the enterohepatic circulation is defined as a circulating BA pool [1,4].”
6. The authors state in their response letter the following: “The most relevant novel findings (i.e., research articles published in the years 2020-2022) have been now cited and discussed in the different sections of the paper.” But simple addition of more recent papers to already well-known facts is not what the Reviewer was asking for. For example, the following sentence now includes two references from 2022, however, at this point they do not contribute anything new to the references from 2020 and 2017: “The enterohepatic circulation of BA is characterized by minimal daily hepatic synthesis and fecal loss. This mechanism requires a tightly organized control of various steps involved in the interaction between microbiota (intestine), BA (liver, gallbladder, intestine), FXR (intestine, liver), fibroblast growth factor 19 (FGF19 in intestine, liver, gallbladder)[3,7,23,24].”
6. Despite including new literature in the other sections of the manuscript, they sometimes appear to be taken out of context and simply listed one after another (like for example references 300 – 305).
7. The underlined phrase is completely unnecessary: “Liver-specific FXR KO mice (characterized by loss of FXR in the liver) do not show protection against diet-induced obesity and develop disturbed glucose homeostasis [183].”
Minor concerns:
1. The authors have to revise the manuscript for some errors, like missing words (examples below), grammar, and some English:
a. Verb missing: “Here, BA as strong detergents for emulsification, solubilization and absorption of dietary fat, cholesterol, and lipid-soluble vitamins.”
b. Preposition missing: “Circulating FGF19 binds the FGFR4/β-Klotho in the gallbladder smooth muscle causing relaxation/refilling, and the FGFR4/β-Klotho the liver activating the FXR-small heterodimer partner (SHP) pathway.”
c. “Several external stimuli interact any time with the intestinal barrier [26] including and nutrients with dietary fiber, proteins, fat, carbohydrates, toxins, chemicals vehiculated by ingested food, water and bile with BA.”
2. Check all abbreviations throughout the manuscript. E.g. FXR is abbreviated already in the introduction but again explained in the second part.
3. Check correct grammar/comma settings/… some examples for wrong or missing commas or hyphen:
a. “A further proof was that Fgf15-knockout mice and intestine-specific Fxr-knockout mice stimulated by FXR agonists, could not repress CYP7A1 [110]”
b. “During fasting glucose is produced after gluconeogenesis and glycogenolysis.”
c. “Fexaramine is the intestine specific FXR agonist that increases FGF15 signaling lead-ing to altered BA pool and increasing the level of the secondary tauro-conjugated BA tauro-litocholic acid (TLCA) [188].”
4. Several formatting errors are present in the MS: E.g. the author contribution section contains several errors (corrected or highlighted in red): “Author Contributions: Conceptualization, A.D. and P.P.; literature review, A.D., L.B., J.B., M. K.; G.G., F.S.; writing—original draft preparation, A.D., J.B., G.G.; writing—review and editing; super-vision, H.W., D.Q.H.W., F.S, P.P.; funding acquisition, P.P. F.S.; has particularly participated to the paragraph on bile acid kinetics. All authors have read and agreed to the published version of the manuscript.”
5. Can the authors clarify in detail which information is new? What are the novel aspects of BA signaling discussed in this review the first time? It still seems that the authors just reviewed previous reviews.
Author Response
Reviewer 1
The manuscript has much improved compared to the initial version and many suggestions of the Reviewers have been addressed or discussed. However, there is still high need for improvement, especially in terms of grammar and the use of some very confusing sentences without much meaning. The abstract contains too much information (it is not necessary to describe all the pathways) and can be shortened. Additionally, the authors have not addressed all of the issues raised, particularly with regard to the correct citation of primary sources and the inclusion of more recent literature (please see comments below).
Answer:
We thank the reviewer for his/her suggestions, which address further points which help us to improve the manuscript. In particular:
- The abstract has been rewritten and shortened (now 222 words).
- The text has been revised for grammar/typo errors to further increase clarity.
- We tried to address all issues raised, also citing primary sources and more recent literature.
- Changes appear in red in the R2 version.
Major concerns:
- The introduction is only citing 3 papers, all published by the corresponding author, ignoring other literature – in particular, no primary sources are cited. More than 70% of the introduction only summarizes summary of a previous review/book chapter, without any primary literature cited.
Answer:
We thank the reviewer for this comment. The introduction section has been now shortened and extensively rewritten. A total of 24 references have been now cited in this section, also including primary literature.
- The following statement on biliary lipids is not correct: “In bile, BA are the major lipid component, followed by phospholipids and almost totally unesterified cholesterol. Overall, the three lipids account for about 99% of total lipids by weight. Bilirubin is another lipid in bile and represents less than 1% of biliary lipids.” The three main lipids of bile are bile salts, phospholipids and cholesterol, which is nonesterified. Free cholesterol accounts for 97% of all sterols in bile, the rest are cholesterol precursors and dietary phytosterols. Additionally, “Almost totally unesterified cholesterol”sounds misleading, as it indicates the existence of partially esterified cholesterol as well, which is incorrect. Please be aware of that. Furthermore, bilirubin is not a lipid species itself and it is present in bile mainly as bilirubin-conjugates.
Answer:
We thank the reviewer for this precise observation. The sentence appears in the introduction and has been changed as follows:
“The three main lipids in bile are bile salts, phospholipids and cholesterol, which is nonesterified. Free cholesterol accounts for 97% of all sterols in bile, the rest are cholesterol precursors and dietary phytosterols”.
- There is no reference indicated for the last paragraph of introduction (describing consequences of disrupted BA homeostasis).
Answer:
The sentence has been now changed as follows:
“Consequences of disrupted BA homeostasis include cholestasis [20], hepatic steatosis, liver fibrosis, and liver tumour [21,22]. Notably, external modulation of the BA-FXR axis is paving the way to innovative and potent therapeutic tools[3,23,24]. ”
- 4.Redundant information: “BA lost in feces daily account for 5% and this amount equals the daily synthesis in the liver (0.2-0.6 g/daily).The enterohepatic circulation of BA consists of continuous recirculation of BA (4-12 times daily) from the intestine to the liver and back to the intestine, depending on meal type and frequency.” (Chapter 2) vs. already described in introduction: “By this dynamic mechanism, BA undergo continuous enterohepatic circulation with 4-12 cycles/day. The fecal loss is minimal at every cycle (5%) and must be compensated by the de novo synthesis of primary BA in the liver (~200-600 mg/daily).”
Answer:
We agree with the reviewer. The sentence in chapter 2 has been deleted.
(“BA lost in feces daily account for 5% and this amount equals the daily synthesis in the liver (0.2-0.6 g/daily). The enterohepatic circulation of BA consists of continuous recirculation of BA (4-12 times daily) from the intestine to the liver and back to the intestine, depending on meal type and frequency.” ).
- Checking literature in the revised manuscript revealed redundant citations. For example, why does the following sentence need two references!? “The cycling of BA between the liver and the intestine is referred to as the enterohepatic circulation, while the total amount of BA in the enterohepatic circulation is defined as a circulating BA pool [1,4].”
Answer:
References have been checked throughout the manuscript, avoiding redundant citations. As a consequence, the number of references has been reduced from 305 (R1 version) to 273 (R2 version).
- The authors state in their response letter the following: “The most relevant novel findings (i.e., research articles published in the years 2020-2022) have been now cited and discussed in the different sections of the paper.”But simple addition of more recent papers to already well-known facts is not what the Reviewer was asking for. For example, the following sentence now includes two references from 2022, however, at this point they do not contribute anything new to the references from 2020 and 2017: “The enterohepatic circulation of BA is characterized by minimal daily hepatic synthesis and fecal loss. This mechanism requires a tightly organized control of various steps involved in the interaction between microbiota (intestine), BA (liver, gallbladder, intestine), FXR (intestine, liver), fibroblast growth factor 19 (FGF19 in intestine, liver, gallbladder)[3,7,23,24].”
Answer:
The section with the cited sentence (and related references) has been rewritten and the manuscript has been extensively revised to limit well-established previous evidence. New findings (i.e., 2020-2022), when available, have been commented throughout the manuscript to avoid a simple addition of citations.
- Despite including new literature in the other sections of the manuscript, they sometimes appear to be taken out of context and simply listed one after another (like for example references 300 – 305).
Answer:
The manuscript has been further shortened and revised to report and comment novelties, without missing relevant evidence from oldest studies. New evidence, in particular, are related to therapeutic aspects (experimental studies and clinical trials), which have been reported and described in dedicated sections of the manuscript avoiding a simple list of citations. We hope that this further revision will be satisfactory for the reviewer and for readers.
- The underlined phrase is completely unnecessary: “Liver-specific FXR KO mice (characterized by loss of FXR in the liver) do not show protection against diet-induced obesity and develop disturbed glucose homeostasis [183].”
Answer:
The underlined sentence has been now deleted from the text.
Minor concerns:
- The authors have to revise the manuscript for some errors, like missing words (examples below), grammar, and some English:
Answer:
The text has been now extensively reviewed for grammar/typo errors.
- Verb missing:“Here, BA as strong detergents for emulsification, solubilization and absorption of dietary fat, cholesterol, and lipid-soluble vitamins.”
Answer:
Abstract. The sentence has been changed as follows: “In the small intestine, BA act as strong detergents for emulsification, solubilization and absorption of dietary fat, cholesterol, and lipid-soluble vitamins.”
- Preposition missing:“Circulating FGF19 binds the FGFR4/β-Klotho in the gallbladder smooth muscle causing relaxation/refilling, and the FGFR4/β-Klotho the liver activating the FXR-small heterodimer partner (SHP) pathway.”
Answer:
Abstract. The sentence has been changed as follows:
Circulating FGF19 to the FGFR4/β-Klotho receptor causes smooth muscle relaxation and refilling of the gallbladder. In the liver the binding activates the FXR-small heterodimer partner (SHP) pathway.
- c.“Several external stimuli interact any time with the intestinal barrier [26] including and nutrients with dietary fiber, proteins, fat, carbohydrates, toxins, chemicals vehiculated by ingested food, water and bile with BA.”
Answer:
Chapter 3. The sentence has been changed as follows:
The gut microbiota is sensitive to dietary habits and nutrients such as dietary fiber, proteins, fat, carbohydrates, [41], and to external toxic agents. These include smoking [42], ethanol consumption [43], and environmental pollutants such as heavy metals and pesticides [44-47]. All these factors can affect the diversity and relative abundance of the gut microbiota [48] during the process of enterohepatic circulation, when the primary BA synthetized and secreted by the liver are transformed into secondary BA in the colon [48,49].
- Check all abbreviations throughout the manuscript. E.g. FXR is abbreviated already in the introduction but again explained in the second part.
Answer:
We thank the reviewer for this comment. All abbreviations have been checked throughout the manuscript.
- Check correct grammar/comma settings/… some examples for wrong or missing commas or hyphen:
Answer:
We thank the reviewer for this comment. The text has been extensively checked for grammar/comma settings.
- “A further proof was that Fgf15-knockout mice and intestine-specific Fxr-knockout mice stimulated by FXR agonists, could not repress CYP7A1 [110]”
Answer:
Chapter 5. The selected text has been changed as follows:
A further proof was that Fgf15 knockout mice and intestine-specific Fxr-knockout mice stimulated by FXR agonists could not repress CYP7A1 [101]
- “During fasting glucose is produced after gluconeogenesis and glycogenolysis.”
Answer:
Chapter 7.2. The selected text has been changed as follows:
During fasting, glucose is produced after gluconeogenesis and glycogenolysis. FGF15/19 release mediates the effects of FXR on glucose and lipid regulation.
- “Fexaramine is the intestine specific FXR agonist that increases FGF15 signaling lead-ing to altered BA pool and increasing the level of the secondary tauro-conjugated BA tauro-litocholic acid (TLCA) [188].”
Answer:
Chapter 7.2. The selected text has been changed as follows:
Fexaramine is the intestine-specific FXR agonist that increases FGF15 signaling, leading to altered BA pool and increasing the level of the secondary tauro-conjugated BA tauro-litocholic acid (TLCA) [162]
- 4.Several formatting errors are present in the MS: E.g. the author contribution section contains several errors (corrected or highlighted in red):“Author Contributions: Conceptualization, A.D.and P.P.; literature review, A.D., L.B., J.B., M. K.; G.G., F.S.; writing—original draft preparation, A.D., J.B., G.G.; writing—review and editing; super-vision, H.W.,Q.H.W., F.S, P.P.; funding acquisition, P.P. F.S.; has particularly participated to the paragraph on bile acid kinetics. All authors have read and agreed to the published version of the manuscript.”
Answer:
The manuscript has been now carefully checked for formatting errors, including the author contribution section.
- Can the authors clarify in detail which information is new? What are the novel aspects of BA signaling discussed in this review the first time? It still seems that the authors just reviewed previous reviews.
Answer:
Thank you for this comment. Please consider that by submitting the first version and then R1, we initially thought that a broad overview of such a large topic was appropriated for the readers of Nutrients. Now, we understand the point of view of this reviewer asking for a shorter version with more updates. In line with this policy, we shortened the abstract and text and extensively revised the manuscript while focusing on the most recent evidence, mainly in terms of translational aspects and possible treatments involving BA signaling pathways. At the same time, we tried to keep the main evidence from prior “historical” and landmark studies, to guide the reader towards a better knowledge of pathogenic and therapeutic pathways. We believe that now the paper discusses new evidence while strengthening old findings. The new evidence relates to therapeutic aspects (experimental studies and clinical trials) involving BA signaling modulation. This topic has a dedicated section in the manuscript. We hope that this further revision will be satisfactory for you, and for the readers, while attracting a considerable number of citations.

Reviewer 3 Report
Overall, the revised manuscript is still in need of very extensive editing to improve readability and clarity.
1. The abstract is way too long and should be shortened to no more than 250 words.
2. Replace “Postcibal” with “postprandial”.
3. The following paragraph is way too long and should be broken down to shorter ones!
The enterohepatic circulation of BA is characterized by minimal daily hepatic syn-thesis and fecal loss. This mechanism requires a tightly organized control of various steps involved in the interaction between microbiota (intestine), BA (liver, gallbladder, intes-tine), FXR (intestine, liver), fibroblast growth factor 19 (FGF19 in intestine, liver, gallblad-der)[3,7,23,24]. In particular, the close interaction between gut microbiota, BA homeostasis and the impact in health and disease is of great interest. The gut is the largest surface in the human body extending for roughly 200-300 m2, at the interface between the external and the internal body environment [25]. Several external stimuli interact any time with the intestinal barrier [26] including and nutrients with dietary fiber, proteins, fat, Nutrients 2021, 13, x FOR PEER REVIEW 4 of 44
carbohydrates, toxins, chemicals vehiculated by ingested food, water and bile with BA. The gut microbiota is a complex and dynamic polymicrobial mixture of hundreds of tril-lions of microorganisms [27] which include bacteria, viruses, fungi (mostly yeasts), bacte-riophages, protists, archaea, and microbial eukariotes [28]. The genes of the microbiota a_r_e_ _l_a_b_e_l_l_e_d_ _a_s_ _“m_i_c_r_o_b_i_o_m_e_” _d_e_s_p_i_t_e_ _t_h_e_ _t_e_r_m_ _m_i_c_r_o_biome is currently defining microbes themselves [29]. The density of microbes is increasing dramatically from 103–104 per gram of gut content in the stomach, to 105–106 in the jejunum, to 108–109 in the terminal ileum, and to 1012-1014 bacteria per gram in the colon. It is estimated that there are over 1,000 bacterial species [30], and about three bacterial cells for every single human body cell [31]. The two major phyla in the human gut are Firmicutes, enriched in Gram-positive bacteria with facultative, anaerobic, bacilli and cocci, and Bacteroidetes consisting of mainly Gram-negative bacteria. The described intestinal ecosystem is very dynamic and resilient to changes, while providing effects at the level of the intestinal barrier which includes the mucin, the enterocytes, the gut immune system, the liver, the bile, the gastric acid, the pancreatic secretions in both health and disease [26,32,33]. In addition, the gut microbiota has the ability to produce a myriad of molecules such which include short-chain fatty ac-ids (SCFAs) [24], p-cresol, p-cresyl-glucuronide (pCG), indoxyl sulphate (IS), indole-3 ace-tic acid (IAA), hydrogen sulphide (H2S), and trimethylamine N-oxide (TMAO). Several end-products including bacterial products such as lypopolisaccharide can translocate to the liver through the portal vein and influence liver functions [23,24,34,35]. Notably, the gut microbiota is essential to transform the primary BA synthetized and secreted by the liver into secondary BA in the colon during the process of enterohepatic circulation [3,7,36]. The steps connected to the biotransformation of primary to secondary BA include deconjugation, oxidation of hydroxyl groups in 3, 7 and 12 positions, and 7-dehydroxyla-tion [37] b_y_ _s_p_e_c_i_f_i_c_ _e_n_z_y_m_e_s_ _s_u_c_h_ _a_s_ _h_y_d_r_o_l_a_s_e_s_ _a_n_d_ _7_α-dehydroxylases. The deconjuga-tion process is systematically performed by Lactobacilli, Bifidobacteria, Clostridium, and Bac-teroidetes [38-41] and catalyzed by the bile salt hydrolase (BSH) which hydrolyzes the am-ide bond between BA and taurine or glycine. This step r_e_l_e_a_s_e_s_ _“f_r_e_e_ _B_A_”._ _Enzymes are a_l_s_o_ _i_n_v_o_l_v_e_d_ _i_n_ _7_α-dehydroxylation, and encoded by BA inducible (bai) genes in anaero-bic Firmicutes, Clostridium clusters IV, XI (Paeniclostridium sordelli), and XIVa (Clostridium scindens, C. hiranonis, C. hylemonae) [37,39]. Metagenomic/metatranscriptomic studies find that bai gene cluster is expressed in most individuals but in a small fraction of total intes-tinal bacteria [42]. The results of the study were extended, and the authors found that only a few metagenome-assembled genomes (MAGs) were linked to Peptostreptococcaceae (Clos-tridium clusters XI) while the majority of MAGs presenting the bai gene cluster were asso-ciated with a Ruminococcaceae clade that still lacks isolates. By contrast, less MAGs be-longed to Lachnospiraceae (Clostridium clusters XIVa) along with eight new isolate contain-ing the bai genes. [43]. BA and gut microbes interact within a continuous bidirectional crosstalk in health but also in disease [26,41,44-47]. Gut dysbiosis can disrupt the BA bio-transformation and the composition of the BA pool, while the increased production of the cytotoxic secondary BA deoxycholic acid (DCA) can damage the integrity of the gut mi-crobiota [48]. BA bind to FXR and this step produces antimicrobial peptides (AMPs) (i.e., angiogenin 1 and RNase family member 4). These AMPs play an active role in inhibiting gut microbial overgrowth and gut barrier dysfunction [49,50]. BA can shape the gut mi-crobiota community by promoting the growth of BA-metabolizing bacteria or inhibiting other bile sensitive bacteria. Gut Eubacterium lentum, Ruminococcus gnavus and Clostridium perfringens decrease the antimicrobial effect of BA via the iso-BA pathway by transforming DCA and LCA into iso-DCA and iso-LCA (3b-OH epimers). The secondary DCA has hy-drophobic and cytotoxic profiles with detergent effects on the bacterial cell membranes and antimicrobial properties [51].
4. FGF19 is the human protein encoded by the FGF19 gene, has endocrine hormonal functions (both have low or no af-finity for heparin sulfate) at a systemic level [97,106].
“Both” refer to which two?
5. Affinity of FGF15/19 is greater for the FGFR4, mainly expressed in the liver than FGFR1, mainly expressed in the white adipose tissue (WAT) der volume in response to meal, by following the rhythmic alternance of emptying-refill-ing episodes.
How come there are two “mainly”? What is der volume and what is rhythmic alternance? As a matter of fact, “alternance” should be alternation.
6. In an animal model of parenteral nutrition-associated cholestasis, the administration of the FXR agonist GW4064 prevented hepatic injury and cholestasis normalizing serum BA, increasing the expression of canalicular bile and of sterol and phospholipid transport-ers, and following a suppression of macrophage recruitment and activation[211].
Rewrite the whole paragraph, which is completely incomprehensible!
7. Excess BA retention implies hepato-cyte damage, steatosis, fibrosis and even liver tumorigenesis [7,218].
What does “imply” mean?
8. Therefore, BA spill over will be evident into the systemic circulation.
No space between spill and over!
9. All acronyms including BSEP should be spelled out in full name in the main text.
10. Another example is that Cyp2c70-/- mice develop a more human-like hydrophobic BA pool, develop liver inflammation [237] and altered FXR signaling [238].
Replace the second “develop” with another verb!
11. This step involves the release of cytochrome c and release of excessive reactive oxygen species (ROS). BA synthesis and conjugation are involved [227,254] and contribute to counteract BA overload in the hepatocytes.
“Counteract” should be “counteracting”!
12. Such molecules are still are able to modulate glucose metabolism but studies are restricted to animal models [263-266] with scarce studies in humans [265,267].
Remove the second “are”!
13. The FGF21 variant LY2405319 has been tested in patients with obesity and type 2 diabetes, with beneficial effects on lipid metabolism, body weight, fasting insulin and adiponectin levels, but no significant reduction in fasting glucose levels [267]. Another long-acting FGF21 variant, PF-05231023, induced in overweight/obese subjects with type 2 diabetes a reduced body weight, an improvement in plasma lipoprotein profile and in adiponectin levels, without effects on blood glucose. In the treated cohort, however, possible effects on bone formation and resorption were noticed[268]. In obese patients with hypertriglyc-eridaemia on atorvastatin, with or without type 2 diabetes, the same molecule reduced triglycerides in the absence of weight loss. In the treated group, however, serious adverse effects were noticed, causing the discontinuation of therapy in some participants[269]. Pegbelfermin (BMS-986036), a PEGylated FGF21 analog has been used in obese patients with type 2 diabetes predisposed to fatty liver, resulting in an improvement of the lipid profile, of fibrosis biomarkers and adiponectin levels, in the presence of mild adverse events[270]. Pegbelfermin has been also used in a phase 2a study in obese/overweight subjects and in NASH patients. In this trial, treatment significantly reduced hepatic fat fraction, in the presence of mild side effects (mainly diarrhea and nausea)[271].
The whole paragraph is about FGF21. Does it have any bearing in BA signaling?
14. w_h_i_l_e_ _i_n_ _β-TC6 cells OCA induces AKT (Protein Kinase B) dependent translocation of glucose transporter 2 (GLUT2).
Add a dash between “AKT” and “dependent”.
15. Tropifexor (LJN452), showed a favourable tolerability and pharmacokinetic profile and showed dose-dependent increase of FGF19 level with no change in serum lipids in healthy volunteers [286].
Again, replace the second showed with another verb!
16. cilofexor treatment was safe and lead to a significant improvement of liver biochemistry and biomarkers of cholestasis and cellular injury.
Replace “lead” with “led”!
17. Future studies should focus on the development of FXR agonists or modulators having beneficial anti-inflammatory effects with poor metabolic actions.
Replace poor with minimal”!
18. (lipid accumulation), liver enzymes (AST, ALT GGT), and bile acids synthesis with
Add “, and” in front of GGT.
Author Response
Reviewer 3
Overall, the revised manuscript is still in need of very extensive editing to improve readability and clarity.
Answer:
We thank the reviewer for his/her suggestions, which contributed to significantly improve the manuscript. The text has been further revised to increase readability and clarity. See also comments directed to Reviewer #1
- The abstract is way too long and should be shortened to no more than 250 words.
Answer:
We thank the reviewer for this suggestion. The abstract has been rewritten and shortened. It now consists of 222 word.
- Replace “Postcibal” with “postprandial”.
Answer:
The term “postcibal” has been replaced with “postprandial” throughout the manuscript.
- The following paragraph is way too long and should be broken down to shorter ones!
The enterohepatic circulation of BA is characterized by minimal daily hepatic syn-thesis and fecal loss. This mechanism requires a tightly organized control of various steps involved in the interaction between microbiota (intestine), BA (liver, gallbladder, intes-tine), FXR (intestine, liver), fibroblast growth factor 19 (FGF19 in intestine, liver, gallblad-der)[3,7,23,24]. In particular, the close interaction between gut microbiota, BA homeostasis and the impact in health and disease is of great interest. The gut is the largest surface in the human body extending for roughly 200-300 m2, at the interface between the external and the internal body environment [25]. Several external stimuli interact any time with the intestinal barrier [26] including and nutrients with dietary fiber, proteins, fat, Nutrients 2021, 13, x FOR PEER REVIEW 4 of 44
carbohydrates, toxins, chemicals vehiculated by ingested food, water and bile with BA. The gut microbiota is a complex and dynamic polymicrobial mixture of hundreds of tril-lions of microorganisms [27] which include bacteria, viruses, fungi (mostly yeasts), bacte-riophages, protists, archaea, and microbial eukariotes [28]. The genes of the microbiota a_r_e_ _l_a_b_e_l_l_e_d_ _a_s_ _“m_i_c_r_o_b_i_o_m_e_” _d_e_s_p_i_t_e_ _t_h_e_ _t_e_r_m_ _m_i_c_r_o_biome is currently defining microbes themselves [29]. The density of microbes is increasing dramatically from 103–104 per gram of gut content in the stomach, to 105–106 in the jejunum, to 108–109 in the terminal ileum, and to 1012-1014 bacteria per gram in the colon. It is estimated that there are over 1,000 bacterial species [30], and about three bacterial cells for every single human body cell [31]. The two major phyla in the human gut are Firmicutes, enriched in Gram-positive bacteria with facultative, anaerobic, bacilli and cocci, and Bacteroidetes consisting of mainly Gram-negative bacteria. The described intestinal ecosystem is very dynamic and resilient to changes, while providing effects at the level of the intestinal barrier which includes the mucin, the enterocytes, the gut immune system, the liver, the bile, the gastric acid, the pancreatic secretions in both health and disease [26,32,33]. In addition, the gut microbiota has the ability to produce a myriad of molecules such which include short-chain fatty ac-ids (SCFAs) [24], p-cresol, p-cresyl-glucuronide (pCG), indoxyl sulphate (IS), indole-3 ace-tic acid (IAA), hydrogen sulphide (H2S), and trimethylamine N-oxide (TMAO). Several end-products including bacterial products such as lypopolisaccharide can translocate to the liver through the portal vein and influence liver functions [23,24,34,35]. Notably, the gut microbiota is essential to transform the primary BA synthetized and secreted by the liver into secondary BA in the colon during the process of enterohepatic circulation [3,7,36]. The steps connected to the biotransformation of primary to secondary BA include deconjugation, oxidation of hydroxyl groups in 3, 7 and 12 positions, and 7-dehydroxyla-tion [37] b_y_ _s_p_e_c_i_f_i_c_ _e_n_z_y_m_e_s_ _s_u_c_h_ _a_s_ _h_y_d_r_o_l_a_s_e_s_ _a_n_d_ _7_α-dehydroxylases. The deconjuga-tion process is systematically performed by Lactobacilli, Bifidobacteria, Clostridium, and Bac-teroidetes [38-41] and catalyzed by the bile salt hydrolase (BSH) which hydrolyzes the am-ide bond between BA and taurine or glycine. This step r_e_l_e_a_s_e_s_ _“f_r_e_e_ _B_A_”._ _Enzymes are a_l_s_o_ _i_n_v_o_l_v_e_d_ _i_n_ _7_α-dehydroxylation, and encoded by BA inducible (bai) genes in anaero-bic Firmicutes, Clostridium clusters IV, XI (Paeniclostridium sordelli), and XIVa (Clostridium scindens, C. hiranonis, C. hylemonae) [37,39]. Metagenomic/metatranscriptomic studies find that bai gene cluster is expressed in most individuals but in a small fraction of total intes-tinal bacteria [42]. The results of the study were extended, and the authors found that only a few metagenome-assembled genomes (MAGs) were linked to Peptostreptococcaceae (Clos-tridium clusters XI) while the majority of MAGs presenting the bai gene cluster were asso-ciated with a Ruminococcaceae clade that still lacks isolates. By contrast, less MAGs be-longed to Lachnospiraceae (Clostridium clusters XIVa) along with eight new isolate contain-ing the bai genes. [43]. BA and gut microbes interact within a continuous bidirectional crosstalk in health but also in disease [26,41,44-47]. Gut dysbiosis can disrupt the BA bio-transformation and the composition of the BA pool, while the increased production of the cytotoxic secondary BA deoxycholic acid (DCA) can damage the integrity of the gut mi-crobiota [48]. BA bind to FXR and this step produces antimicrobial peptides (AMPs) (i.e., angiogenin 1 and RNase family member 4). These AMPs play an active role in inhibiting gut microbial overgrowth and gut barrier dysfunction [49,50]. BA can shape the gut mi-crobiota community by promoting the growth of BA-metabolizing bacteria or inhibiting other bile sensitive bacteria. Gut Eubacterium lentum, Ruminococcus gnavus and Clostridium perfringens decrease the antimicrobial effect of BA via the iso-BA pathway by transforming DCA and LCA into iso-DCA and iso-LCA (3b-OH epimers). The secondary DCA has hy-drophobic and cytotoxic profiles with detergent effects on the bacterial cell membranes and antimicrobial properties [51].
Answer:
We agree with the reviewer. In Chapter 3 the paragraph has been now extensively rewritten and shortened, reducing the text from 755 to 258 word:
The gut represents a dynamic interface between the external and the internal body environment [39] and several stimuli interact continuously with the intestinal barrier [40]. The gut microbiota is sensitive to dietary habits and nutrients such as dietary fiber, proteins, fat, carbohydrates, [41], and to external toxic agents. These include smoking [42], ethanol consumption [43], and environmental pollutants such as heavy metals and pesticides [44-47]. All these factors can affect the diversity and relative abundance of the gut microbiota [48] during the process of enterohepatic circulation, when the primary BA synthetized and secreted by the liver are transformed into secondary BA in the colon [48,49]. BA and gut microbes interact within a continuous bidirectional crosstalk in health but also in disease [40,50,51]. Gut dysbiosis can disrupt the BA biotransformation and the composition of the BA pool, while the increased production of the cytotoxic secondary BA deoxycholic acid (DCA) can damage the integrity of the gut microbiota [9]. BA bind to FXR and this step produces antimicrobial peptides (AMPs) (i.e., angiogenin 1 and RNase family member 4). These AMPs play an active role in inhibiting gut microbial overgrowth and gut barrier dysfunction [52]. BA can also shape the gut microbiota community by promoting the growth of BA-metabolizing bacteria or inhibiting other bile sensitive bacteria. Gut Eubacterium lentum, Ruminococcus gnavus and Clostridium perfringens decrease the antimicrobial effect of BA via the iso-BA pathway by transforming DCA and LCA into iso-DCA and iso-LCA (3b-OH epimers). The secondary DCA has hydrophobic and cytotoxic profiles with detergent effects on the bacterial cell membranes and antimicrobial properties [53].
- FGF19 is the human protein encoded by the FGF19 gene, has endocrine hormonal functions (bothhave low or no af-finity for heparin sulfate) at a systemic level [97,106].
“Both” refer to which two?
Answer:
We thank the reviewer for this precise observation. There was a mistake in the composition of the sentence. In chapter 5 the sentence is:
FGF19 is the human protein encoded by the FGF19 gene and has endocrine hormonal functions at a systemic level [88,96].
- Affinity of FGF15/19 is greater for the FGFR4, mainlyexpressed in the liver than FGFR1, mainly expressed in the white adipose tissue (WAT) der volume in response to meal, by following the rhythmic alternance of emptying-refilling episodes.
How come there are two “mainly”? What is der volume and what is rhythmic alternance? As a matter of fact, “alternance” should be alternation.
Answer:
Chapter 5. The sentence has been rewritten as follows:
Affinity of FGF15/19 is greater for the FGFR4, which is mainly expressed in the liver, than for FGFR1, primarily expressed in the white adipose tissue (WAT) [109,110]
- The term “alternance” has been replaced with “alternation”:
This phenomenon is well visible during the ultrasonographic functional study of time-dependent changes of gallbladder volume in response to meal, by following the rhythmic alternation of emptying-refilling episodes [115,117-119].
- In an animal model of parenteral nutrition-associated cholestasis, the administration of the FXR agonist GW4064 prevented hepatic injury and cholestasis normalizing serum BA, increasing the expression of canalicular bile and of sterol and phospholipid transport-ers, and following a suppression of macrophage recruitment and activation[211].
Rewrite the whole paragraph, which is completely incomprehensible!
Answer:
Thank you.
Chapter 8.1. The paragraph has been rewritten as follows:
In an animal model of parenteral-nutrition-induced cholestasis, the administration of the FXR agonist GW4064 prevented hepatic injury and cholestasis. Treated animals showed a normalization of serum BA levels which were associated with increased expression of canalicular bile, of sterol and phospholipid transporters, and with suppression of macrophage recruitment and activation. These effects were secondary to the restoration of hepatic FXR signaling [184].
- Excess BA retention implieshepato-cyte damage, steatosis, fibrosis and even liver tumorigenesis [7,218].
What does “imply” mean?
Answer:
Chapter 8.2 The term has been replaced with “generates”:
Excess BA retention generates hepatocyte damage, steatosis, fibrosis and even liver tumorigenesis [21,22].
- Therefore, BA spill overwill be evident into the systemic circulation.
No space between spill and over!
Answer:
Chapter 8.1. The sentence has been corrected:
Therefore, BA spillover will be evident into the systemic circulation, as confirmed in both animal and human models [19,189,195-198].
- All acronyms including BSEP should be spelled out in full name in the main text.
Answer:
The acronym BSEP has been spelled out in full name (bile salt export pump) in the main text
- Another example is that Cyp2c70-/- mice developa more human-like hydrophobic BA pool, develop liver inflammation [237] and altered FXR signaling [238].
Replace the second “develop” with another verb!
Answer:
Chapter 8.2. The sentence has been changed as follows:
“Another example is that Cyp2c70-/- mice develop a more human-like hydrophobic BA pool with liver inflammation [225] and altered FXR signaling [226]”
- This step involves the release of cytochrome c and release of excessive reactive oxygen species (ROS). BA synthesis and conjugation are involved [227,254] and contribute to counteractBA overload in the hepatocytes.
“Counteract” should be “counteracting”!
Answer:
We thank the reviewer. Chapter 8.2. The sentence has been changed as suggested:
Both basolateral and canalicular BA transporters and enzymes governing BA synthesis and conjugation are involved [195,218] and contribute to counteracting BA overload in the hepatocytes [219].
- Such molecules are still areable to modulate glucose metabolism but studies are restricted to animal models [263-266] with scarce studies in humans [265,267].
Remove the second “are”!
Answer:
We thank the reviewer. Chapter 9.1. The sentence has been changed as suggested:
Such molecules are still able to modulate glucose metabolism but studies are restricted to animal models [228-230] with scarce studies in humans [229,231].
- The FGF21 variant LY2405319 has been tested in patients with obesity and type 2 diabetes, with beneficial effects on lipid metabolism, body weight, fasting insulin and adiponectin levels, but no significant reduction in fasting glucose levels [267]. Another long-acting FGF21 variant, PF-05231023, induced in overweight/obese subjects with type 2 diabetes a reduced body weight, an improvement in plasma lipoprotein profile and in adiponectin levels, without effects on blood glucose. In the treated cohort, however, possible effects on bone formation and resorption were noticed[268]. In obese patients with hypertriglyc-eridaemia on atorvastatin, with or without type 2 diabetes, the same molecule reduced triglycerides in the absence of weight loss. In the treated group, however, serious adverse effects were noticed, causing the discontinuation of therapy in some participants[269]. Pegbelfermin (BMS-986036), a PEGylated FGF21 analog has been used in obese patients with type 2 diabetes predisposed to fatty liver, resulting in an improvement of the lipid profile, of fibrosis biomarkers and adiponectin levels, in the presence of mild adverse events[270]. Pegbelfermin has been also used in a phase 2a study in obese/overweight subjects and in NASH patients. In this trial, treatment significantly reduced hepatic fat fraction, in the presence of mild side effects (mainly diarrhea and nausea)[271].
- The whole paragraph is about FGF21. Does it have any bearing in BA signaling?
Answer:
We sincerely thank the reviewer for this comment, which allowed us to better explain the relationships between FGF21 and BA synthesis/signaling. The section 9.1 (now titled “FGF19 and FGF21 variants) has been changed as follows:
“9.1. FGF19 and FGF21 variants
Few studies have focused on the role of FGF19 variants devoid of stimulatory effect on FGFR4 (because of potential tumorigenic activity). Mimetic molecules include chimeric molecules FGF19-4, 5 and 6 (mutagenesis in the N-terminus and in the heparin binding domains in amino acids 38–42), FGF19v (Conjugation of amino acids 1–20 of FGF21 with amino acids 25–194 of FGF19), M70 (3 amino acid substitutions and 5 amino acid deletions in the N-terminus). Such molecules are still able to modulate glucose metabolism but studies are restricted to animal models [228-230] with scarce studies in humans [229,231]. The translational value of FGF19 variants is still under evaluation and further evidence is awaited in this field.
FGF21 belong to the FGF19 subfamily of endocrine FGFs and, as FGF19, require the co-receptor β Klotho for binding and signaling through the FGF receptors[232]. As shown in experimental models, FGF21 is a negative regulator of BA synthesis, being able to decrease BA levels in the liver and in the small intestine, with a significant reduction of the BA pool size. These findings are paralleled by decreased colonic and fecal BA, with a concomitant increase in fecal cholesterol and fatty acid excretions [233]. The modulatory effect of FGF21 on BA synthesis seems independent of the FXR/FGF15 pathway [234]. Due to beneficial metabolic effects, FGF21 variants are emerging as promising therapeutic tools in metabolic diseases. The FGF21 variant LY2405319 has been tested in patients with obesity and type 2 diabetes, with beneficial effects on lipid metabolism, body weight, fasting insulin and adiponectin levels, but no significant reduction in fasting glucose levels [231]. Another long-acting FGF21 variant, PF-05231023, induced in overweight/obese subjects with type 2 diabetes a reduced body weight, an improvement in plasma lipoprotein profile and in adiponectin levels, without effects on blood glucose. In the treated cohort, however, possible effects on bone formation and resorption were noticed[235]. In obese patients with hypertriglyceridaemia on atorvastatin, with or without type 2 diabetes, the same molecule reduced triglycerides in the absence of weight loss. In the treated group, however, serious adverse effects were noticed, causing the discontinuation of therapy in some participants[236]. Pegbelfermin (BMS-986036), a PEGylated FGF21 analog has been used in obese patients with type 2 diabetes predisposed to fatty liver, resulting in an improvement of the lipid profile, of fibrosis biomarkers and adiponectin levels, in the presence of mild adverse events[237]. Pegbelfermin has been also used in a phase 2a study in obese/overweight subjects and in NASH patients. In this trial, treatment significantly reduced hepatic fat fraction, in the presence of mild side effects (mainly diarrhea and nausea)[238]. Of note, Pegbelfermin promotes a significant reduction from baseline in serum concentrations of DCA and conjugates in patients with NASH and in overweight/obese adults, with a potential role in modulating secondary BA synthesis also by gut microbiome [239]. The extent of total BA reduction recorded in the cited study (about 20-30%)[239] is comparable to that obtained following treatment with FXR agonists[240], with the advantage to be selective for secondary BA[239].
- w_h_i_l_e_ _i_n_ _β-TC6 cells OCA induces AKT (Protein Kinase B) dependenttranslocation of glucose transporter 2 (GLUT2).
Add a dash between “AKT” and “dependent”.
Answer:
We thank the reviewer. Chapter 9.2. The sentence has been changed as suggested:
“…OCA induces AKT (Protein Kinase B)-dependent translocation of glucose transporter 2 (GLUT2)
- Tropifexor (LJN452), showeda favourable tolerability and pharmacokinetic profile and showed dose-dependent increase of FGF19 level with no change in serum lipids in healthy volunteers [286].
Again, replace the second showed with another verb!
Answer:
We thank the reviewer. Chapter 9.2 The sentence has been changed as follows:
Tropifexor (LJN452) showed a favourable tolerability and pharmacokinetic profile and induced a dose-dependent increase of FGF19 level, with no change in serum lipids in healthy volunteers [254].
- cilofexor treatment was safe and leadto a significant improvement of liver biochemistry and biomarkers of cholestasis and cellular injury.
Replace “lead” with “led”!
Answer:
Chapter 9.2. The sentence has been changed as follows:
“…cilofexor treatment was safe and led to a significant..”
- Future studies should focus on the development of FXR agonists or modulators having beneficial anti-inflammatory effects with poor metabolic actions.
Replace poor with minimal”!
Answer:
We thank the reviewer. The sentence has been changed as suggested:
Research needs to develop FXR agonists or modulators with beneficial anti-inflammatory effects and minimal metabolic actions
- (lipid accumulation), liver enzymes (AST, ALT GGT), and bile acids synthesis with
Add “, and” in front of GGT.
Answer:
We thank the reviewer. The sentence has been changed as suggested:
“… liver enzymes (AST, ALT, and GGT), and..”

Reviewer 4 Report
The authors satisfactorily addressed the raised issues.
There are only a few minor corrections I suggest.
1. In page 2, CA, CDCA, LCA, and DCA were used first time. Please identify the full name of the BAs.
2. In the Fig. 1, (III) is probably meant to be tertiary BA. What is the exact meaning of tertiary BA (probably mentioning UDCA)? If you want to keep it in the graphic, please indicate in the legend.
Author Response
Reviewer 4
The authors satisfactorily addressed the raised issues.
There are only a few minor corrections I suggest.
- In page 2, CA, CDCA, LCA, and DCA were used first time. Please identify the full name of the BAs.
Answer:
We thank the reviewer for this suggestion. The sentence has been changed as follows:
Both primary and secondary BA activate FXR with the following rank order: chenodeoxycholic acid (CDCA) > lithocholic acid (LCA) = deoxycholic acid (DCA) > cholic acid (CA) in the conjugated and unconjugated forms
- In the Fig. 1, (III) is probably meant to be tertiary BA. What is the exact meaning of tertiary BA (probably mentioning UDCA)? If you want to keep it in the graphic, please indicate in the legend.
Answer:
The legend of Figure 1 has been changed as follows:
Figure 1. Summary of key factors involved in the composition of the bile acid pool. Several dynamic events are active daily either in the fasting and postprandial period (grey area) and contribute to shape the qualitative/quantitative profile of the bile acid pool, its maintenance and expansion. Such events work in concert with bile acid synthesis and related steps (orange box). Abbreviations: FGF-19, fibroblast growth factor 19; FXR, farnesoid X receptor; SHP, small heterodimer partner; primary bile acids are cholic and chenodeoxycholic acid; secondary bile acids are deoxycholic and lithocholic acids; tertiary bile acid is ursodeoxycholic acid.
